# Lsm7 phase-separated condensates trigger stress granule formation

Michelle Lindström[1,7], Lihua Chen [1,2,7,8 ✉], Shan Jiang [1], Dan Zhang[1], Yuan Gao[1], Ju Zheng[3], Xinxin Hao [1], Xiaoxue Yang[1], Arpitha Kabbinale[1], Johannes Thoma [1,4], Lisa C. Metzger [1,4], Deyuan Y. Zhang [5], Xuefeng Zhu[6], Huisheng Liu [2], Claes M. Gustafsson [6], Björn M. Burmann [1,4], Joris Winderickx [3], Per Sunnerhagen [1] & Beidong Liu [1,8 ✉]

Stress granules (SGs) are non-membranous organelles facilitating stress responses and linking the pathology of age-related diseases. In a genome-wide imaging-based phenomic screen, we identify Pab1 co-localizing proteins under 2-deoxy-D-glucose (2-DG) induced stress in *Saccharomyces cerevisiae*. We find that deletion of one of the Pab1 co-localizing proteins, Lsm7, leads to a significant decrease in SG formation. Under 2-DG stress, Lsm7 rapidly forms foci that assist in SG formation. The Lsm7 foci form via liquid-liquid phase separation, and the intrinsically disordered region and the hydrophobic clusters within the Lsm7 sequence are the internal driving forces in promoting Lsm7 phase separation. The dynamic Lsm7 phase-separated condensates appear to work as seeding scaffolds, promoting Pab1 demixing and subsequent SG initiation, seemingly mediated by RNA interactions. The SG initiation mechanism, via Lsm7 phase separation, identified in this work provides valuable clues for understanding the mechanisms underlying SG formation and SG-associated human diseases.

[1] Department of Chemistry and Molecular Biology, University of Gothenburg, Gothenburg, Sweden. [2] Guangzhou Laboratory, Guangzhou, Guangdong, China. [3] Functional Biology, KU Leuven, Leuven, Belgium. [4] Wallenberg Centre for Molecular and Translational Medicine, University of Gothenburg, Gothenburg, Sweden. [5] College of Artificial Intelligence, Shenyang Aerospace University, Shenbei New District, Shenyang, Liaoning, China. [6] Department of Medical Biochemistry and Cell Biology, University of Gothenburg, Gothenburg, Sweden. [7] These authors contributed equally: Michelle Lindström, Lihua Chen. [8] These authors jointly supervised this work: Lihua Chen, Beidong Liu. ✉email: chen_lihua@gzlab.ac.cn; beidong.liu@cmb.gu.se

Cytoplasmic stress granules (SGs) are non-membranous organelles with a dynamic structure that form transiently to reprogram RNA translation under stress conditions by affecting mRNA function and localization[1–6]. SGs typically contain substantial quantities of non-translating mRNAs, translation initiation components, and additional proteins affecting mRNA function[1,4,7]. In addition, cellular signaling factors and catalytic proteins have been shown to be sequestered in SGs[2,8,9]. Therefore, SGs can facilitate cellular responses and promote cell survival under stress conditions. However, excessive formation and persistence of SGs has been implicated as an underlying causative event in neurodegenerative diseases and cancer progression[10–13].

Several efforts have been made to identify SG-associated proteins. Using protein cross-linking coupled immunoprecipitation, the Parker lab has identified the major mRNA-binding proteins under glucose deprivation and analyzed the relocation of these proteins during stress, including assembly into SGs and processing bodies (PBs)[14]. In a subsequent study, the Parker lab analyzed the proteome of the G3BP1-associated SG stable core, and identified numerous new SG constituents in mammalian cells[4]. Since SGs are actively exchanging materials with the cytosol, and given that the composition of SGs varies depending on the type of stress[15,16], it remains important to investigate the SG composition and the interactions seen with typical SG marker proteins.

The poly(A)-binding protein (Pab1 in budding yeast) is highly conserved across eukaryotes and has been shown to be consistently present in SGs under various stressful conditions[1,15,17]. Recent studies have highlighted its role in various stress responses. Indeed, overexpressing Pab1 in yeast improves robustness against various stressors including oxidative stress, heat, and acetic acid[18]. Pab1 also serves as a stress sensor and forms phase separation hydrogels to promote organism fitness under stress conditions[19]. These observations encouraged us to explore which proteins co-localize with Pab1 under stressful conditions and understand the mechanism whereby these proteins modulate SG formation.

The Lsm7 protein belongs to the conserved Lsm1-7/Pat1 complex, some components of which have been shown to co-localize to PBs in yeast (Pat1 and Lsm1)[20] and human cells (Lsm1 and Lsm4)[21,22]. As other Sm-like proteins, Lsm7 plays a role in RNA metabolism including the mRNA 5'-to-3' decay pathway[23] and pre-mRNA splicing[24]. It is still uncertain if Lsm7 possesses the same properties as other Sm-like proteins that are recruited into PBs or are essential for PB formation. Since SGs and PBs share mRNPs and actively shuttle material between them[25,26], it is of high interest to determine whether Lsm7 is also co-localized with SGs, and if so, whether it plays an active role in SG formation.

Here, we apply an imaging-based phenomic screen to search for proteins co-localizing with Pab1-RFP (C-terminally tagged red fluorescent protein) under 2-deoxy-D-glucose (2-DG, glycolysis inhibitor) induced stress conditions[27]. We show that Lsm7 forms foci co-localizing with SGs and that these foci are dynamic liquid-like phase-separated condensates promoting Pab1 demixing and further SG initiation.

## Results

### A global survey identifies proteins co-localizing with SGs under 2-DG
To identify proteins that are recruited into SGs under 2-DG induced stress conditions, we performed a genome-wide phenomic screen using the yeast GFP-tagged protein collection[28] to identify SG components that co-localize with the marker protein Pab1-RFP (Fig. 1a). Co-localization refers to an observed overlap between two different fluorescent labels (see Methods). Proteins for which the co-localization was more

than 60% were defined as strongly SG-residing. Using this definition, we found 14 yeast proteins that strongly localized to the SGs under 2-DG treatment (Fig. 1b). Among these were known PB components (Ssd1, Dcp1, and Dcp2 (Supplementary Fig. 1a)) and SG components (Pub1, eIF4E, Tif4632 (eIF4G2), Tif4631 (eIF4G1), and Nab6), as well as previously identified PB/SG-shared components (Nam7, Hek2, Sbp1)[14,29,30]. In addition, we identified previously unreported Pab1 co-localizing proteins, such as Lsm7, Iki3, and Nst1 (Fig. 1b, d and Supplementary Fig. 1b). To provide an overall view on all hits identified from the screen, a complete screen list of Pab1 co-localizing proteins, including the proteins below the 60% co-localization threshold, is included in Supplementary Data 3.

Next, we performed a gene network analysis and found that these SG co-localizing proteins form a dense interaction network through physical (and genetic) interactions (Fig. 1c), thereby further supporting their co-localization with Pab1. This analysis defined the poly (A) + RNA-binding protein Pub1 and the hnRNPK-like protein Hek2 as central nodes besides Pab1, interacting with many of the other SG components. This suggests that both proteins have important roles in modulating PB and SG assembly under 2-DG induced stress conditions[31].

Among the identified proteins, Lsm7, a component of the Lsm1-7/Pat1 complex, showed a strong co-localization (87.4 ± 3.1%) with Pab1-RFP granules under 2-DG stress (Fig. 1d). The Lsm7 protein expression level was unaffected by the 2-DG treatment (Fig. 1e). Interestingly, Lsm7 has been reported as one of the most stable proteins in the cell with a protein half-life above 100 h, much longer as compared to the other Lsm1-7/Pat1 complex components (9.1–16.7 h)[32]. This raises the question whether Lsm7 has other functions in addition to being a subunit of the Lsm1-7/Pat1 complex. Since Lsm7 forms barely any foci under non-stress condition (Supplementary Fig. 3a, top panel), we wanted to further confirm that the close proximity relationship between Lsm7 and Pab1 is specifically induced by the 2-DG treatment. We performed a proximity ligation assay (PLA)[33,34] to monitor the proximity localization between Pab1 and Lsm7 with or without 2-DG treatment. We found a clear PLA signal in the Lsm7-5xFlag strain under 2-DG treatment (Fig. 1f, middle) but no signal under the non-treated condition (Fig. 1f, bottom). This confirms that Lsm7 is in close proximity to the SG marker Pab1 under 2-DG treatment.

### The Lsm7 effects on SGs are partially independent of its role in PBs
Previously published data showed indications of a possible link between Lsm7, PBs, and SGs[20,35,36]. First, the Lsm1-7 complex has been reported as a PB component[20]. Second, it was shown that Lsm7 can modulate the toxicity of human FUS (FUsed in Sarcoma), a pathogenic protein that assembles into SGs under stress conditions[35] and known to be associated with human amyotrophic lateral sclerosis[37]. Third, some of the Lsm1-7 complex components have been shown to modulate stress responses through binding to stress-activated mRNAs[36], a phenomenon that is suggested to be implicated in tumor progression[38] as well as virus replication and infection[5,39,40]. Hence, to confirm the link between Lsm7 and SGs, we first determined whether Lsm7 is needed for SG formation. We observed that the deletion of *LSM7* led to a significant reduction in SG formation under 2-DG treatment, as indicated by Pab1-RFP granules (Fig. 2a) and Pbp1-GFP granules (Supplementary Fig. 2a). In line with this observation, SG formation was increased in an *LSM7* overexpression strain (Fig. 2b and Supplementary Fig. 2b), without affecting Pab1 or Pbp1 protein expression levels (Fig. 2c and Supplementary Fig. 2c). SGs are known to form under different stress conditions[5]. Therefore, we

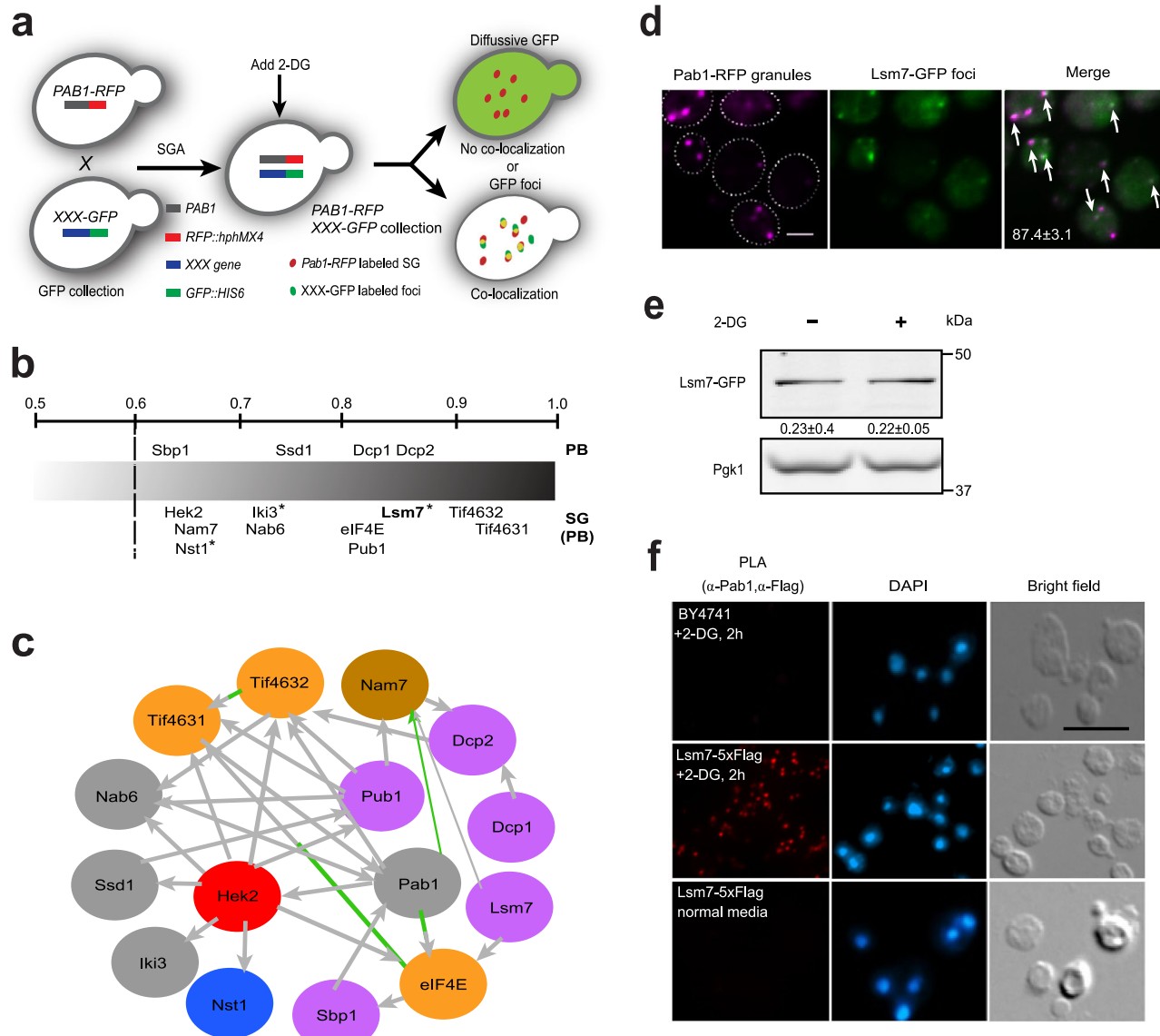

**Fig. 1 A global survey identifies Lsm7 as a SG component. a** Work flow for the SG components screen under 2-DG treatment. **b** Fractional co-localization of GFP foci with Pab1-RFP granules based on manual fluorescence microscopic studies. Proteins that are known PB components are indicated above the gradient, and proteins that are reported as SG components (or PBs) are shown below the gradient. Previously unreported SG components are marked with an asterisk. Numbers above the gradient represent co-localization rate (%) of GFP foci with Pab1-RFP granules. Four biologically independent experiments were examined and >200 cells were analyzed for each (mean ± S.D). **c** Interaction network analysis for the 14 hits that co-localize with SGs together with Pab1 (colored nodes are colored based on their enrichment of Gene Ontology biological processes; gray for metabolism, pink for RNA processing; orange for protein biosynthesis, blue for stress response, brown for DNA metabolism, and red for RNA localization; gray lines indicate physical interactions, and green lines indicate genetic interactions). **d** Pab1-RFP granules were strongly co-localized with Lsm7-GFP foci (arrow heads indicate the co-localization). Values represent the co-localization rate (%). Scale bar indicates 2 μm. Three biologically independent experiments were examined and >200 cells were analyzed for each (mean ± S.D). **e** The Lsm7-GFP protein expression level was not impacted by the addition of 2-DG. Log-phase cells were treated with 400 mM 2-DG for 2 h and subsequently collected for Western blot analysis. Data are representative of three biologically independent experiments. Values are means ± S.D of the arbitrary units (intensity of target bands normalized to Pgk1 levels) for each clone. **f** Lsm7 proximately associates with Pab1 under 2-DG treatment. WT (BY4741) and Lsm7-5xFlag strains were grown and treated with or without 2-DG. In situ proximity, ligation assay was performed by using antibodies against Pab1 and Flag-tag. The PLA signal was assessed as described in Methods. Scale bar indicates 5 μm. Three biologically independent experiments were examined and representative images from one experiment are shown. Source data are provided as a Source Data file.

decided to investigate the impact of *LSM7* deletion on SG formation with other known stressors (Fig. 2d). Lsm7 also impacts SG formation under glucose starvation (-glu 2 h) and stationary phase, but not during sodium azide stress ($NaN_3$) and heat shock, indicating that Lsm7 might play a greater role during nutrient and glucose stress regulation than the other two tested stresses (Fig. 2d).

Our findings implicate Lsm7 in SG regulation; however, the composition of Pab1-linked Lsm7 foci, and whether Lsm7 functions as part of the Lsm1-7 complex in P-bodies, remains unclear. We decided to focus on possible candidates of the complex components and a known associating protein Pat1. All candidates display significantly lower levels of foci co-localization with Pab1 granules under 2-DG stress than the co-localization

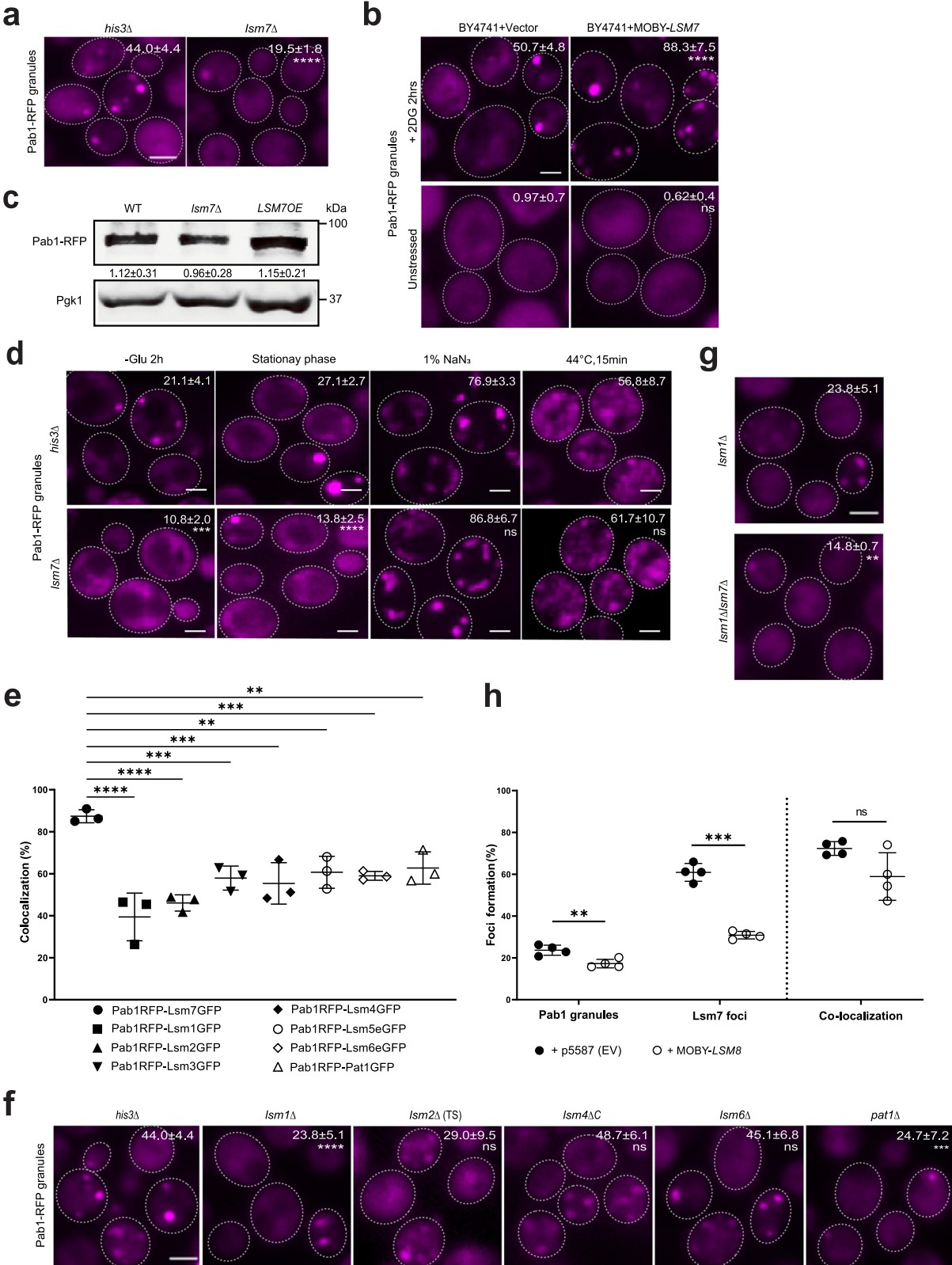

between Lsm7 foci and Pab1 granules, indicating that some free Lsm7 co-localizes with Pab1 outside of the Lsm1-7 complex residing within PBs (Fig. 2e) and that the Lsm7 effects on SG formation could be partially independent of its role as a PB component.

The Pab1 2-DG stress phenotype was tested for the available null strains of the Lsm1-7/Pat1 complex components. The other deletion mutants do not show the same effects as the *lsm7Δ* mutant in Pab1 granule formation (especially for *lsm2Δ*, *lsm4ΔC*, and *lsm6Δ* mutants), suggesting that the *lsm7Δ* effects on SGs are not limited to Lsm7's function in the Lsm1-7/Pat1 complex (Fig. 2f and a). However, we noticed that there is a statistically significant decrease in Pab1 granule formation in the *lsm1Δ* and *pat1Δ* strains as compared to the WT (*his3Δ*). Furthermore,

**Fig. 2 Lsm7 effects on SG formation.** Scale bar indicates 2 μm. If not stated otherwise, stress was induced by 2 h 2-DG treatment and >200 cells were analyzed for each biologically independent experiment. Values represent percentage of foci formation or co-localization and are shown as mean ± S.D. **a** Deletion of *LSM7* decreases SG formation as compared to the control (*his3Δ*). Seven biologically independent experiments were examined (unpaired two-tailed *t*-test). ****$p < 0.0001$. **b** Overexpression of *LSM7* increases SG formation in the WT (BY4741) strain as compared to empty vector control. No SG induction under unstressed conditions. Six biologically independent experiments were examined (unpaired two-tailed *t*-test). ****$p < 0.0001$, ns = 0.3317. **c** The Pab1 protein expression levels were not changed in the *lsm7Δ* or the *LSM7OE* strain. Data are representative of three biologically independent experiments. **d** Deletion of *LSM7* affects SG formation under glucose starvation and stationary phase, as compared to the control (*his3Δ*). Six biologically independent experiments were examined and >300 cells were analyzed for each (unpaired two-tailed t-test). Left to right: ***$p = 0.0003$, ****$p < 0.0001$, ns = 0.0519, ns = 0.4126. **e** Lsm7 foci co-localize with SGs to a greater extent than other Lsm1-7/Pat1 complex components. Three biologically independent experiments were examined (one-way ANOVA followed by Dunnett's test). Left to right: ****$p < 0.0001$, ***$p = 0.0007$, ***$p = 0.0003$, **$p = 0.0018$, ***$p = 0.001$, **$p = 0.0036$. **f** Deletion of some Lsm1-7/Pat1 components results in decreased SG formation. Seven (*his3Δ*, *lsm1Δ*, *pat1Δ*), six (*lsm6Δ*), five (*lsm4ΔC*), and three (*lsm2Δ*(TS)) biologically independent experiments were examined (one-way ANOVA followed by Dunnett's T3 multiple comparisons test, with individual variances computed for each comparison). Left to right: ****$p < 0.0001$, ns = 0.3105, ns = 0.5875, ns=0.9983, ***$p = 0.0006$. **g** Deletion of both *LSM7* and *LSM1* results in greater loss of SGs than the *lsm1Δ* single deletion. Seven (*lsm1Δ*) and three (*lsm1Δlsm7Δ*) biologically independent experiments were examined (unpaired two-tailed *t* test with Welch's correction). **$p = 0.003$. **h** Overexpression of *LSM8* affects Lsm7 foci and SG formation as compared to empty vector control. Four biologically independent experiments were examined (unpaired two-tailed *t* test). Left to right: **$p = 0.007$, ***$p = 0.0002$, ns = 0.096. Source data are provided as a Source Data file.

deletion of *LSM1* or *PAT1* clearly impacts Lsm7 foci formation as well (Supplementary Fig. 2d and e). However, the Lsm7 foci that remain in *lsm1Δ* still co-localize with Pab1 granules to a high degree (49 ± 17%, Supplementary Fig. 2d). The effects on SGs and PBs in the *pat1Δ* mutant have been reported before[41]. The marked decrease in SGs in the *pat1Δ* mutant cannot be fully explained by its effect on PBs (minor decrease in PBs[41]), indicating other mechanisms partaking in Pat1-related SG regulation. Accordingly, deletion of either *PAT1* or *LSM1* results in predominantly nuclear localization of Lsm7 under 2-DG stress, indicating that these proteins affect Lsm7 cellular localization and subsequent Lsm7 ability to form cytoplasmic foci and promotion of SG formation (Supplementary Fig. 2d). Moreover, the double-mutant *lsm1Δ lsm7Δ* displays an even stronger loss of SGs (Fig. 2g), as compared to the single mutants (Fig. 2a and g), indicating an additive effect of Lsm7 on SG formation.

Further, the effects of deletion of the C-terminal tail of Lsm4, which has a prion-type domain that contributes to aggregation of P bodies[42], were tested. The *lsm4ΔC* strain does not show an impact on SG formation under 2-DG stress (Fig. 2f). This strain has previously been shown to affect PB and SG formation under glucose depletion[41]. However, 2-DG is a weaker stressor, resulting in glucose limitation, not complete depletion. It has been shown that the SG composition and regulation vary between different stressors (reviewed in ref. [43]), then perhaps *lsm4ΔC*-related SG effects require more severe stress conditions to emerge.

It is known that Lsm8 can influence the cellular location (cytoplasmic or nuclear) of Lsm7 and other Lsm1-7/2-8-complex components[44,45]. Therefore, we overexpressed *LSM8*, which is supposed to increase the fraction of Lsm7 accumulated in the nucleus, to investigate the effects on Lsm7 foci and SGs. We observed that there is a statistically significant decrease in Lsm7 foci formation and SG formation when overexpressing *LSM8* during 2-DG stress (Fig. 2h). Similarly, we observed that there is an increased nuclear localization of Lsm7 in *LSM1* or *PAT1* deletion strains. Moreover, a significant decrease in Lsm7 foci formation and a subsequent decrease in SG formation was also observed in these mutants (Supplementary Fig. 2d and e). These results show that alterations in the proteins contributing to the balancing of the cellular localization of Lsm7 can alter Lsm7 cytoplasmic foci formation and subsequent SG promotion function.

**Lsm7 impacts SG formation without affecting PB formation.** Given that some components of the Lsm1-7/Pat1 complex have been suggested as PB components in yeast (Pat1 and Lsm1)[20] and

human cells (Lsm1 and Lsm4)[21,22], and that PBs promote SG formation under glucose deprivation[41], it is possible that the decreased SG formation in *lsm7Δ* is simply due to defective PB formation caused by *LSM7* deletion. Therefore, we also tested whether PB formation is decreased in the *lsm7Δ* strain. By using Dcp2-GFP as a PB marker, we found that *LSM7* deletion did not influence PB formation or signal intensity of PBs (Fig. 3a and b). This result further supports Lsm7's independent function in SG formation. Moreover, under normal glucose conditions, Lsm7 does not form foci, unlike Dcp2 (Supplementary Fig. 3a). However, we do observe that the number of Lsm7 foci during 2-DG stress is decreased when PB formation is hampered by deleting the PB components *EDC3* or *DCP2* (temperature-sensitive allele) (Supplementary Fig. 3b and c), indicating that PBs (or PB components) could partly associate with and enhance Lsm7 foci formation and further promote SG formation. It has been shown that deletion of *DCP2* results in an increase in PB formation under glucose depletion[30]. However, differences in strain and stressor might influence the complex Dcp2 protein interaction patterns and subsequent PB phenotype, possibly explaining the different phenotypes.

To further elucidate the relationship between PBs, Lsm7, and SGs, we overexpressed *LSM7* in a PB (Dcp2-GFP) and SG (Pab1-RFP) deficient deletion mutant (*edc3Δ*) to see if exogenous Lsm7 could rescue the phenotype (Fig. 3c). There was no effect on the PB formation; however, overexpression of *LSM7* in this mutant did result in a statistically significant increase in SGs, suggesting that Lsm7 can affect SG formation without the need for increased PB formation (Fig. 3c).

Although, as reported by previous studies[41], PBs have a function in SG formation, we found evidence indicating that specifically, Lsm7 may have functions in SG formation independent from its role as a PB component. First, Lsm7 does not display the foci formation phenotype typical of PBs under unstressed conditions (Supplementary Fig. 3), neither do Lsm7 foci grow in size during prolonged stress, as reported for Dcp2[46] (Fig. 4e, f). Second, deletion of *LSM7* does not impact on PBs (Fig. 3a) but does result in decreased SG formation (Fig. 2a). Lastly, when overexpressing *LSM7* in a SG and PB deficient mutant there is a significant increase in SG formation and no significant effect on PBs (Fig. 3c).

**Lsm7 foci formation is needed to promote SGs.** To further clarify the mechanisms of how Lsm7 influences SG formation, we first determined if Lsm7 modulates Pab1 protein expression, as Pab1 is known to promote SG formation under glucose

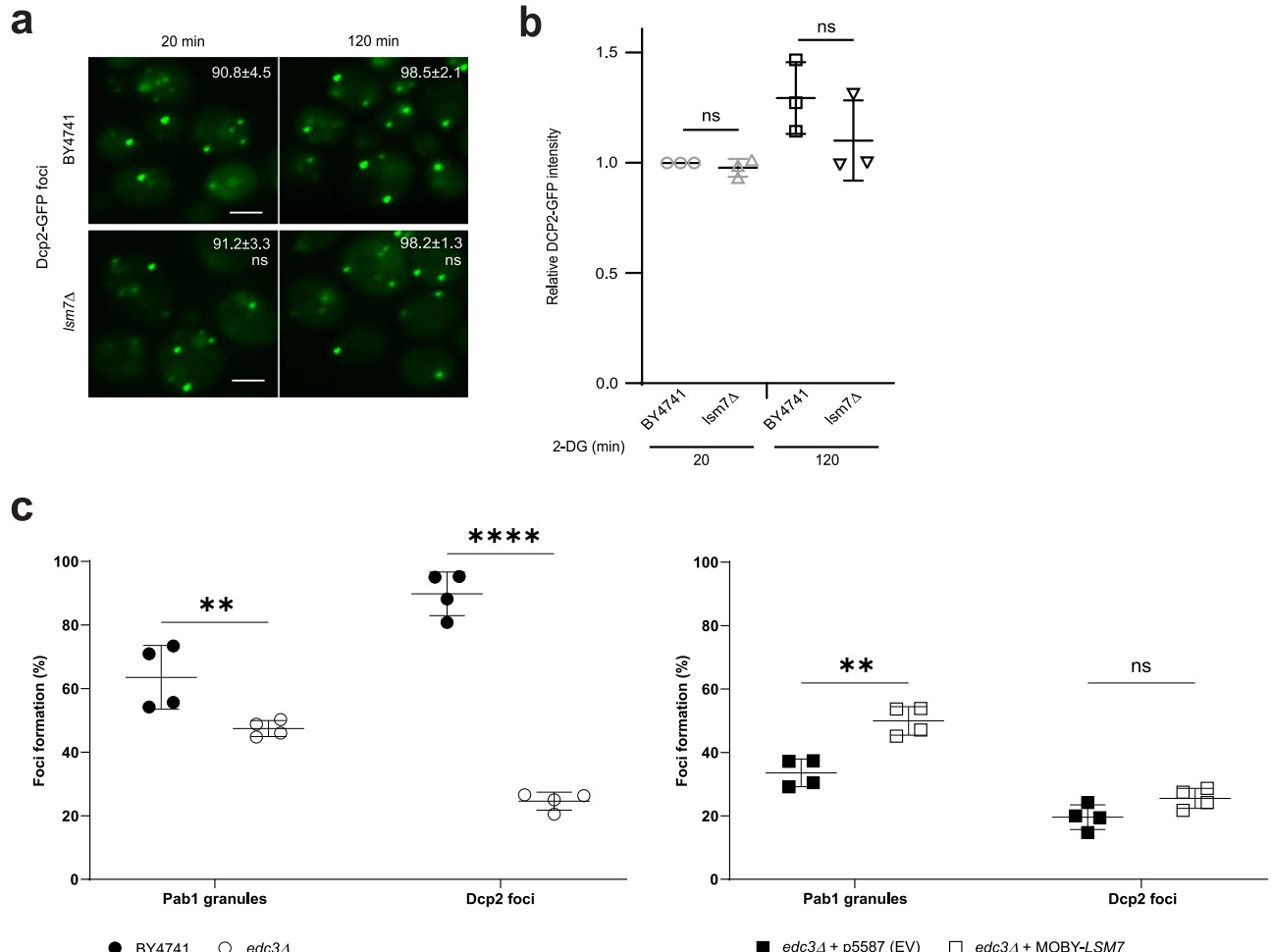

**Fig. 3 Lsm7 impacts SG formation without affecting PB formation. a** PB formation is not significantly changed in the *lsm7Δ* mutant. WT (BY4741) and *lsm7Δ* mutant strains were treated with 2-DG for 20 min and 120 min, followed by sample collection and imaging. Values represent percentage of cells with Dcp2-GFP foci and are shown as mean ± S.D. Scale bar indicates 2 μm. Six biologically independent experiments were examined and >300 cells were analyzed for each (unpaired two-tailed *t* test). Left to right: ns = 0.8732, ns = 0.7548. **b** There is no statistically significant difference in the relative Dcp2-GFP signal intensity for the WT (BY4741) and the *lsm7Δ* mutant. Individual value points are shown with mean ± S.D. Three biologically independent experiments were examined for each strain and one representative image for each was analyzed for GFP signal (unpaired two-tailed *t* test). Left to right: ns = 0.1189, ns = 0.6956. **c** Overexpression of *LSM7* can increase the 2-DG-induced SG formation in the SG- and PB-deficient *edc3Δ* mutant. Deletion of *EDC3* results in a decrease in SGs (Pab1-RFP) and PBs (Dcp2-GFP), as compared to the WT (BY4741) (left). The SG phenotype can be partially rescued by overexpression of *LSM7*, without affecting the number of PBs, as compared to the empty vector control (right). Individual value points are shown with mean ± S.D of percentage of cells with Pab1-RFP granules and Dcp2-GFP foci. Four biologically independent experiments were examined and >200 cells were analyzed for each (two-way ANOVA followed by Tukey's test). Left to right: **p = 0.0014, ****p < 0.0001, **p = 0.0011, ns = 0.4091. Source data are provided as a Source Data file.

deprivation[47]. Our aforementioned results showed that Pab1 protein levels were not affected in either *lsm7Δ* or *LSM7* overexpression strains (Fig. 2c). Therefore, we speculated that Lsm7 foci formation might be the factor that affects SG formation. Recent studies suggest that the formation of membrane-less compartments is driven by a physical process called phase separation[48–50]. SGs have been suggested to form via a multistep process, which is facilitated by liquid-liquid phase separation (LLPS) of SG-associated components[6,51–53]. We thus asked whether the SG-associated Lsm7 foci are one of the early phase separation components involved in the formation of SGs. By using multiple predictive algorithms (Fig. 4a), we identified that Lsm7 possesses two intrinsically disordered regions (IDRs), the presence of which have been shown to mediate protein phase separation in several studies[54–57] (Fig. 4a, top panel). Lsm7 does not seem to have prion-like domains (PLDs) (Fig. 4a, second panel), which are thought to promote phase separation as

well[58–60]. However, there are studies indicating IDRs could drive phase separation without the existence of PLDs[61,62]. We further analyzed the Lsm7 sequence using a yeast LLPS predictor database[63–65]. The prediction shows that Lsm7 has a positive propensity to phase separate near the N-terminal (Fig. 4a, third panel), which overlaps with its IDR region. Hydrophobic interactions have also been implicated as driving factors in LLPS and subsequent protein aggregation[66,67]. There are two hydrophobic regions (aa 39-53 and aa 90-103) in Lsm7 as predicted by ProtScale (Fig. 4a, bottom panel).

These prediction results indicate that Lsm7 could be a potential phase separating protein. To test this hypothesis, we first constructed three truncated lsm7 mutants (*lsm7ΔIDR*-GFP, *lsm7Δ39-53*-GFP, and *lsm7Δ90-103*-GFP), in which the first and most prominent IDR or two segments of the hydrophobic clusters of Lsm7 were deleted (Fig. 4a, Supplementary Fig. 4a). In these mutants, both the number of Lsm7-GFP foci (Fig. 4b, upper

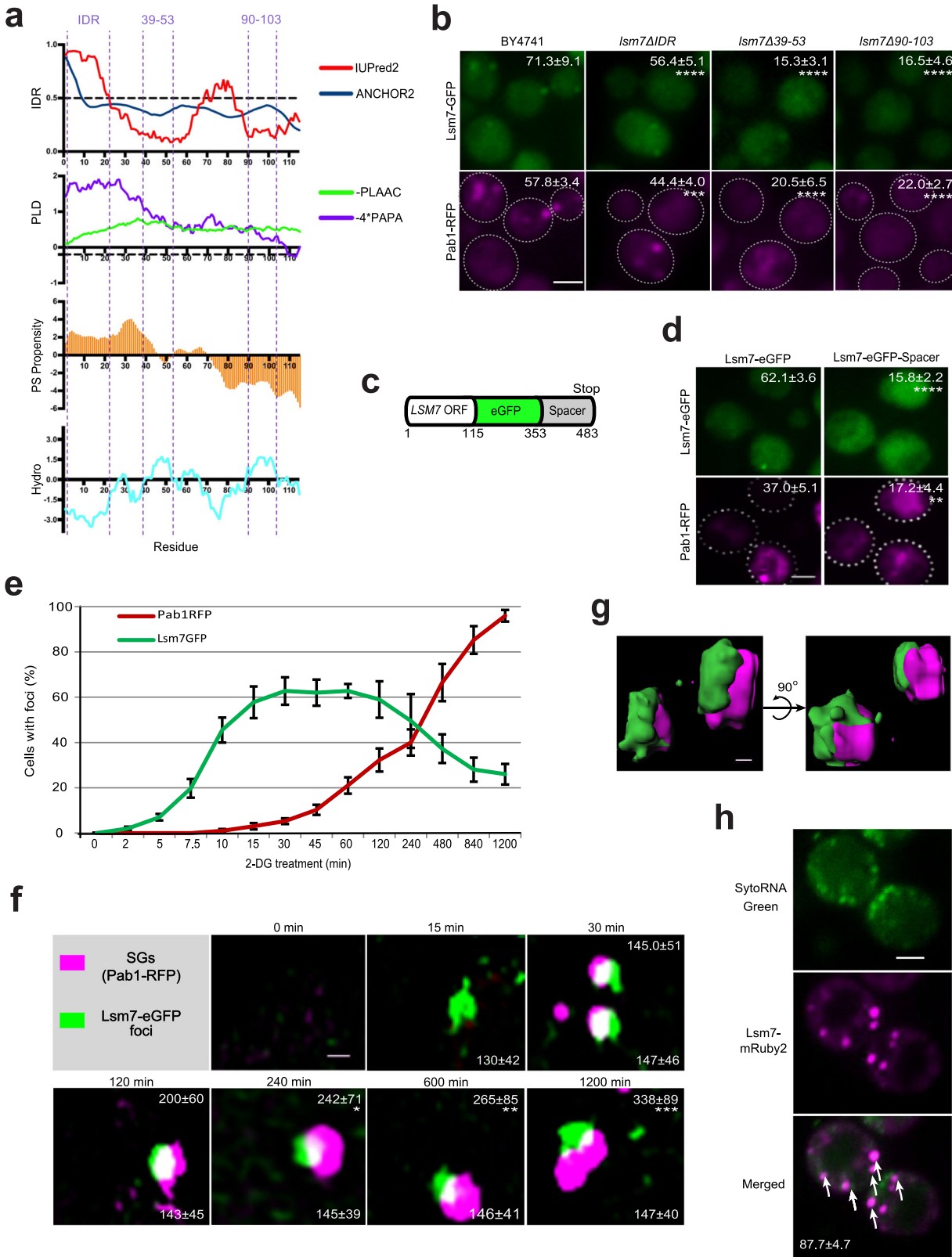

panel) and SGs (Fig. 4b, lower panel) were significantly decreased under 2-DG stress. Hence, to further confirm the role of Lsm7 foci in the SG assembly machinery, we constructed a Lsm7 strain carrying a spacer sequence (the charged middle domain (M) of Sup35)[68] at the C-terminus of Lsm7-eGFP (Fig. 4c). The charged M domain confers solubility to the yeast prion protein Sup35 and has been shown to dampen protein aggregation when fused to

other proteins[68–70]. We wondered if, fusing the spacer to Lsm7, it could alter the weak interactions required for LLPS and make the protein more soluble. Results showed that Lsm7 foci with this modified construct were significantly reduced (Fig. 4d, upper panel) and also SG formation was significantly decreased (Fig. 4d, lower panel). The Lsm7 protein expression level was not affected by the Lsm7 truncations or the fusion to spacer domain, as

**Fig. 4 Lsm7 foci formation is needed to promote SG formation.** If not stated otherwise, stress was induced by 2 h 2-DG treatment and scale bar indicates 2 μm. Values represent percentage of foci formation or co-localization and are shown as mean ± S.D. If not stated otherwise, >300 cells were analyzed for each biologically independent experiment. **a** Bioinformatic analysis of Lsm7's phase separation potential. IDR; intrinsically disordered region prediction, IUPred2 (red, >0.5 regarded as IDR), ANCHOR2 (blue, >0.5 regarded as disordered binding region). PLD; prion-like domains prediction (PLAAC, green, <0 predicted as prion-like; PAPA, purple, <−0.02 or below the dashed line predicted as prion-like). PS propensity; LLPS propensity prediction (>0 treated as positive propensity). Hydro; hydrophobicity prediction (Kyte & Doolittle, >0 treated as hydrophobic). Lsm7 domains of interest are highlighted with dashed lines (IDR, amino acids 39-53 and 90-103). **b** Lsm7-GFP mutants exhibited significantly decreased Lsm7 foci (top) and SG formation (bottom) as compared to the WT (BY4741). Seven (BY4741, lsm7Δ39-53) and six (lsm7ΔIDR, lsm7Δ90-103) biologically independent experiments were examined and >200 cells were analyzed for each (two-way ANOVA followed by Dunnett's multiple comparisons test, with individual variances computed for each comparison). ****$p < 0.0001$, ***$p = 0.0001$. **c** Construction of LSM7-eGFP-spacer strain. **d** Lsm7 foci (top) and SG formation (bottom) are significantly reduced in the spacer-tagged strain. Six biologically independent experiments were examined (unpaired two-tailed t-test). ****$p < 0.0001$, **$p = 0.0014$. **e** Lsm7 foci appear much earlier than SGs. Four biologically independent experiments were examined. **f** 3D-SIM shows the development process and co-localization structures of the Lsm7 foci and SGs. Scale bar indicates 250 nm. Values represent diameters (nm) of SGs (top) and Lsm7 foci (bottom) (mean ± S.D). Representative data for four biologically independent experiments (one-way ANOVA followed by Dunnett's test, compared to the corresponding signal of the time-point 30 min). *$p = 0.047$, **$p = 0.0089$, ***$p = 0.00021$. **g** 3D-surface construction of Lsm7-eGFP and Pab1-RFP foci signals. Scale bar indicates 100 nm. **h** Lsm7 foci harbor RNA. White arrows indicate co-localizing RNA and mRuby2 foci. Representative images from three biologically independent experiments. Source data are provided as a Source Data file.

compared to WT (Supplementary Fig. 4b). Thus, our combined results indicate that the foci formation of Lsm7 plays a role in SG formation under 2-DG treatment.

To obtain an in-depth understanding of the mechanism by which Lsm7 foci formation influences SG formation, we performed a time-course study with super-resolution three-dimensional structured illumination microscopy (3D-SIM). We found that Lsm7 foci appeared already 2 min after 2-DG addition, while the Pab1-RFP granules only started to form about 15 to 30 min later (Fig. 4e). The number of Lsm7 foci and SGs increased with time (Fig. 4e) and reached maximal values at 15 min for Lsm7 foci (~60% of all cells had Lsm7 foci) and at 14 h for SGs (~95% of all cells had SGs) (Fig. 4e). We also found that SGs began to increase in size once formed (Fig. 4f, upper values), while the size of the Lsm7 foci remained about the same throughout the whole period (Fig. 4f, lower values). After a careful alignment for 3D-SIM (Supplementary Fig. 4c), the 3D surface reconstruction results of Lsm7-eGFP and Pab1-RFP (Fig. 4g and Supplementary Movie 1) showed that Lsm7 foci and SGs did not completely overlap with each other. Rather, it appeared that SGs localized on the side of the Lsm7 foci. The microscopy results showed that the Lsm7 foci may seed or work as nucleation sites for cells to build up SGs when encountering 2-DG stress.

SGs have been reported to harbor RNA species[46,71]. To confirm whether Lsm7 foci also co-localize with RNA in vivo, we used a Syto RNASelect Green probe[72] to visualize cellular RNA and Lsm7-mRuby2 to visualize the foci. Our results show that the Lsm7 foci induced by 2-DG co-localize with RNA (Fig. 4h). This suggests that 2-DG-induced Lsm7 foci associate with RNA species as has been shown for SGs[15,26,73]. SGs are known to require non-translating mRNAs for their formation[41]. Hence, we decided to test whether access to non-translating mRNA is important for Lsm7 foci formation as well. When pretreated with cycloheximide (CHX), which traps mRNA on polysomes, Lsm7-eGFP cannot form foci under 2-DG stress (Supplementary Fig. 4d). Furthermore, the Lsm7-eGFP foci disassembly is facilitated by CHX (Supplementary Fig. 4e), in accordance with what has been shown for SGs[41]. When treating a puromycin-sensitized triple mutant (pdr1Δ pdr3Δ snq2Δ) with 2-DG and puromycin, we see an increased induction in Lsm7-eGFP foci formation (Supplementary Fig. 4f), as compared to 2-DG treatment alone. This implies that puromycin, through its ability to dissociate polysomes, can enhance Lsm7 foci formation under 2-DG stress, in accordance with what has been shown for SGs[73].

**Lsm7 foci are liquid–liquid phase-separated condensates**. Our data indicate that foci formation of Lsm7 contributes to SG formation in vivo (Fig. 4b and e). Next, we wanted to determine whether Lsm7 can undergo phase separation in vitro. We observed that purified Lsm7 protein could form phase-separated spherical condensates in the presence of the crowding agent Dextran 70, as well as Ficoll 400 (Fig. 5a and Supplementary Fig. 5). The number and size of these condensates rose with increasing Lsm7 protein concentration (Fig. 5b). The Lsm7 condensates displayed a sensitivity to pH, even in the absence of a crowding agent (Fig. 5c). Higher concentrations of salt negatively affected the demixing properties of Lsm7, suggesting that electrostatic interactions might mediate phase separation of Lsm7[74] (Fig. 5c). Pub1 and Pab1 have shown similar tendencies to demix in vitro under low pH and inability to demix at higher salt concentrations[19,74]. The Lsm7 condensate morphology underwent a distinct change at pH 5.5 giving rise to branched clusters of spherical droplets (Fig. 5c). These condensates visually resemble the in vitro condensates formed by Pub1 and Pab1 under heat shock and pH 6.5[19,74].

We observed that deletion of the predicted phase separation-linked Lsm7 domains resulted in decreased Lsm7 foci formation in vivo (Fig. 4b). Will deletion of these domains result in reduced Lsm7 droplet formation in vitro as well? We found that while WT Lsm7 can still form phase-separated droplets (10% Dextran 70, 5 μM protein concentration), the mutants that lack the IDR region or the aa 90–103 hydrophobic cluster do not display phase separation droplet formation (Fig. 5d). This indicates that these domains could be the internal driving forces in promoting Lsm7 LLPS, thereby explaining the in vivo Lsm7 foci formation phenotypes observed for these mutants (Fig. 4b).

As purified Pab1 has been reported to phase separate in vitro[19], we wondered whether this phase separation is affected by the presence of Lsm7. We found that both purified Pab1 and Lsm7 phase separate in vitro (Fig. 6a). When added together at the same concentration, Pab1 and Lsm7 co-phase separate into the same droplets (Fig. 6b). Furthermore, the addition of 5 μM Lsm7 can increase Pab1 phase separation droplet formation at low concentration (0.5 μM) (Fig. 6c), implying that co-phase separation of these two proteins can result in an enhanced effect, even without the addition of other components. Emerging evidence suggests that intermolecular RNA interactions can directly promote the assembly of RNP granules, contributing to the hypothesis that RNP granule formation might be the combined result of protein–protein, protein–RNA, and RNA–RNA interactions[71]. We see that upon addition of total RNA to

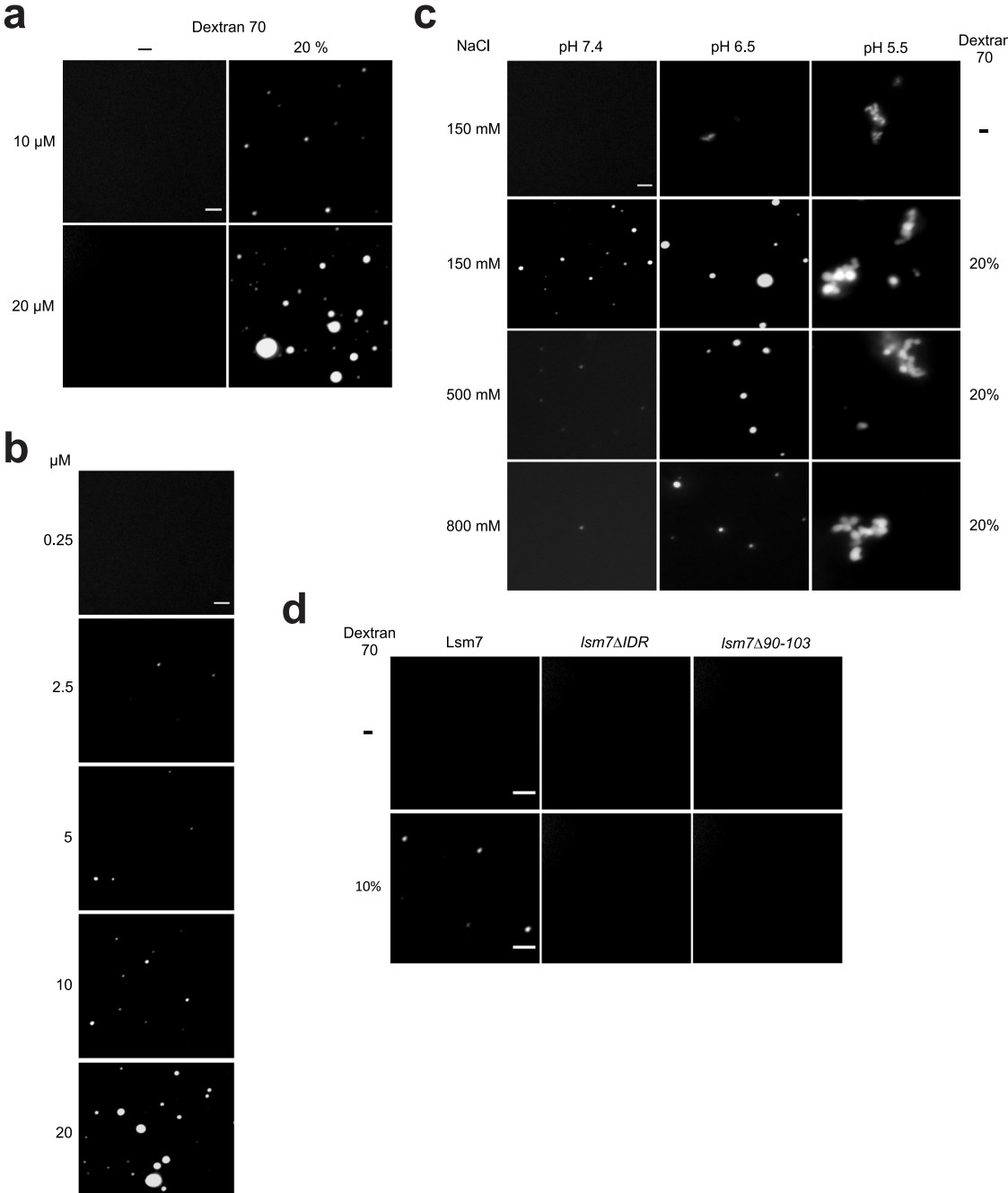

**Fig. 5 Lsm7 undergoes phase separation in vitro. a** Lsm7-GFP was analyzed for the formation of condensates at room temperature with or without the addition of 20% Dextran 70 (pH 7.4, NaCl 150 mM). Images show representative data from three independent experiments. Scale bar, 2 μm. **b** Lsm7-GFP phase-separated condensates are concentration-dependent (pH 7.4, NaCl 150 mM, 20% Dextran 70). Images show representative data from three independent experiments. Scale bar, 2 μm. **c** Lsm7-GFP condensates (10 μM) are pH and salt sensitive. 20% Dextran 70 was added to all except for the first row. Images show representative data from three independent experiments. Scale bar, 2 μm. **d** Lsm7-GFP, *lsm7ΔIDR*-GFP and *lsm7Δ90-103*-GFP (5 μM) were analyzed for the formation of condensates with or without the addition of 10% Dextran 70 (pH 7.4, NaCl 150 mM). Images show representative data from three independent experiments. Scale bar, 2 μm.

purified Lsm7, the droplet formation of Lsm7 is increased (Fig. 6d). Addition of total RNA to Pab1 alone results in loss of demixing (Fig. 6d), an in vitro Pab1 phenotype that has been reported before under heat shock stress[19]. However, addition of Lsm7 has the ability to rescue the phase separation droplet formation of Pab1 in the presence of total RNA (Fig. 6e). It has been proposed that Pab1 needs to release RNA in order to demix[19]. Accordingly, the rescue of Pab1 demixing by Lsm7, in the presence of RNA, might be explained by a replacement of

Pab1–RNA interactions with enhanced Lsm7–RNA interactions and/or inter-protein interactions.

The chemical 1,6-hexanediol, together with the permeabilizing chemical digitonin, has been suggested to dissolve dynamic liquid-like phase-separated assemblies, but not solid-like assemblies[75,76]. The full impact of 1,6-hexanediol on phase-separated assemblies is not completely clear since some reports suggest that 1,6-hexanediol can induce stress and subsequent SG formation under certain conditions and time-points[6]. However,

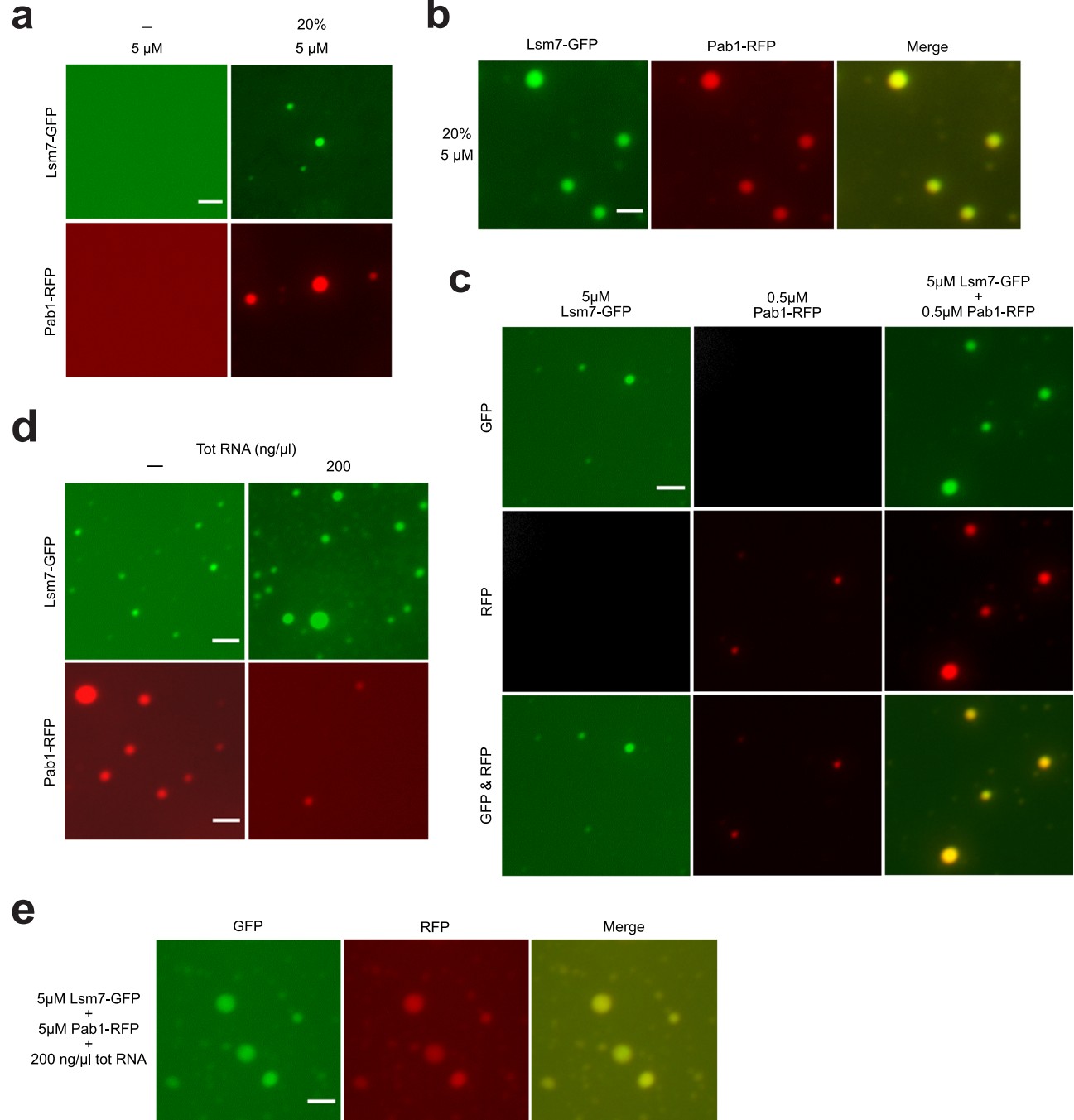

**Fig. 6 Lsm7 co-phase separates with Pab1 in vitro. a** Purified Lsm7-GFP and Pab1-RFP phase separate in vitro with the addition of 20% Dextran 70 (pH 7.4, NaCl 150 mM, protein concentration 5 μM). Images show representative data from three independent experiments. Scale bar, 2 μm. **b** Lsm7-GFP and Pab1-RFP co-phase separate when added together (20% Dextran 70, pH 7.4, NaCl 150 mM, protein concentration 5 μM). Images show representative data from three independent experiments. Scale bar, 2 μm. **c** The addition of 0.5 μM Pab1 together with 5 μM Lsm7 results in an increased droplet formation (20% Dextran 70, pH 7.4, NaCl 150 mM). Images show representative data from three independent experiments. Scale bar, 2 μm. **d** Addition of total yeast RNA (200 ng/μl) enhances Lsm7-GFP droplet formation and decreases Pab1-RFP demixing (20% Dextran 70, pH 7.4, NaCl 150 mM, protein concentration 5 μM). Images show representative data from three independent experiments. Scale bar, 2 μm. **e** During co-phase separation of Lsm7 and Pab1 (5 μM), in the presence of total RNA (200 ng/μl), Lsm7 can rescue the demixing of Pab1 (20% Dextran 70, pH 7.4, NaCl 150 mM). Images show representative data from three independent experiments. Scale bar, 2 μm.

this chemical could provide useful indications of a protein assembly's physical state, when used with forethought and caution. We set out to utilize 1,6-hexanediol and digitonin to test the in vivo physical states of the Lsm7 foci and SGs formed under 2-DG stress. Our data showed that 1,6-hexanediol could dissolve the Lsm7 foci formed under 2-DG treatment but did not

affect the Pab1 granules (Fig. 7a). Furthermore, the number of Pab1 granules was not significantly higher than in the digitonin control, indicating that 1,6-hexanediol treatment does not induce a clear additional stress under the specific condition used here (Fig. 7a). The different effects of 1,6-hexanediol on Lsm7 foci and Pab1 granules might indicate that Lsm7 and Pab1 have different

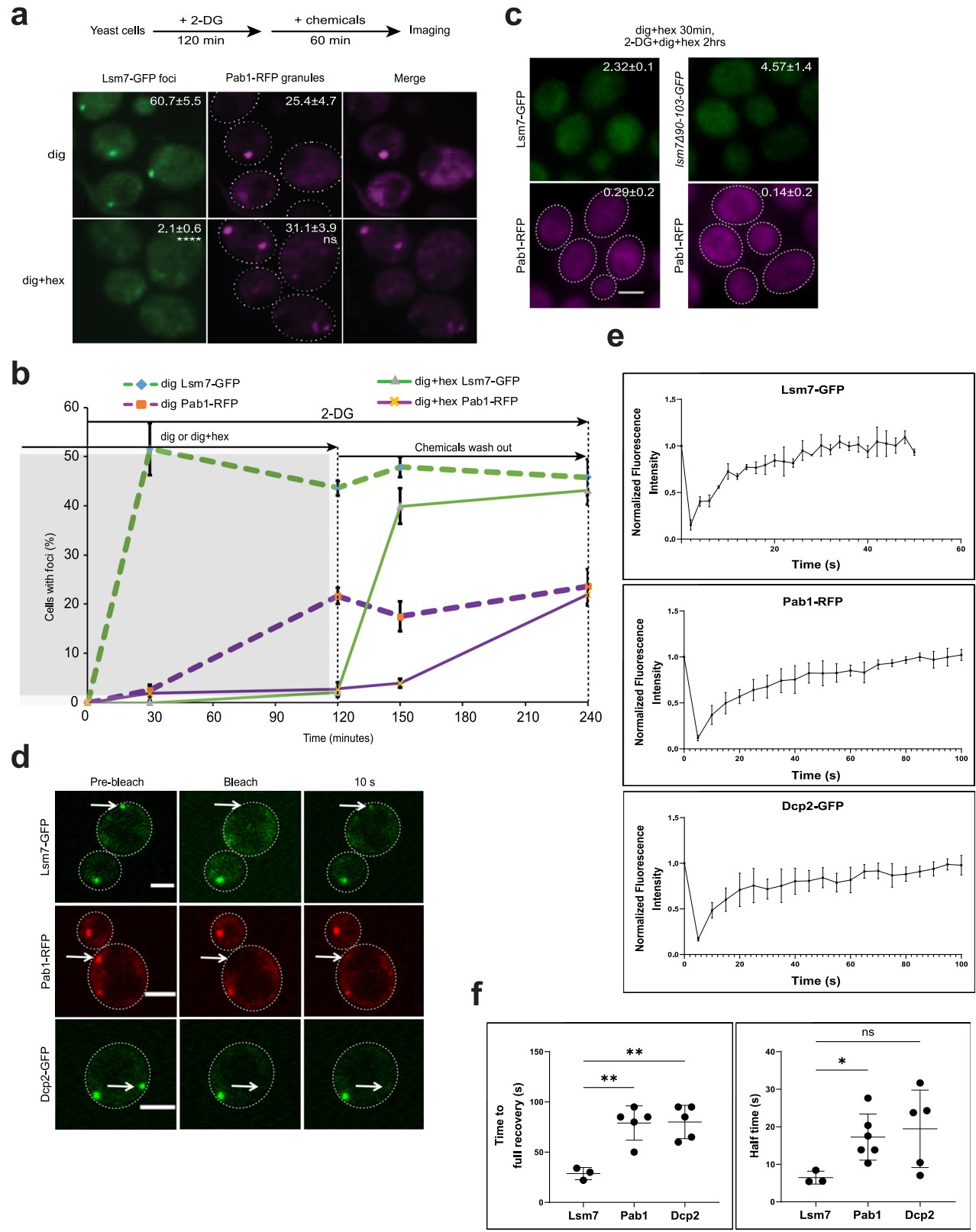

condensate dynamics (Fig. 7a). Moreover, the observation that Pab1 granules, induced by 2-DG, seem to be unaffected by the addition of 1,6-hexanediol, indicates that Lsm7 condensates are not needed for the maintenance of preformed Pab1 granules (Fig. 7a). However, this does not exclude a requirement of Lsm7 foci formation for the initial Pab1 granule formation. Therefore, we pretreated the WT and a *LSM7* truncated mutant strain

(*lsm7Δ90-103*) with 1,6-hexanediol, and noted that neither Lsm7 nor Pab1 granule formation was induced upon 2-DG treatment (Fig. 7b and c), indicating that Lsm7 foci formation might need to occur in order for Pab1 granule formation to initiate. Furthermore, once 1,6-hexanediol was washed out from the media, Lsm7 condensates formed promptly while Pab1 granules formed afterward (Fig. 7b) as observed before (Fig. 4e).

**Fig. 7 Lsm7 phase-separated condensates are dynamic and promote SG formation under 2-DG treatment.** Scale bar indicates 2 μm and values represent mean ± S.D. **a** 1,6-hexanediol could dissolve the Lsm7 foci formed under 2-DG treatment but did not seemingly affect the Pab1 granules (dig, digitonin; dig +hex, digitonin plus 1,6-hexanediol). Values represent percentage of cells with Lsm7 foci or SGs. Four biologically independent experiments were examined and >200 cells were analyzed for each (unpaired two-tailed t-test). ****$p < 0.0001$, ns = 0.1063. **b** 1,6-hexanediol could block 2-DG-induced Lsm7 foci or Pab1 granule formation. (left, gray area). Once 1,6-hexanediol was washed out from the media, Lsm7 foci formed promptly while Pab1 granules formed afterward (right). Four biologically independent experiments were examined and >200 cells were analyzed for each. **c** Dissolved foci phenotype of Pab1-RFP and Lsm7-GFP pre-treated with 1,6-hexanediol and digitonin (30 min) followed by addition of 2-DG for 2 h. The SG and Lsm7 foci formation was hampered in the WT Lsm7-GFP (left) strain as well as the *lsm7Δ90-103*-GFP mutant (right). Three biologically independent experiments were examined and >200 cells were analyzed per clone. Values represent percentage of cells with Lsm7 foci or SGs. **d** Images depicting in vivo FRAP analysis on Lsm7-GFP, Pab1-RFP, and Dcp2-GFP foci after 2 h 2-DG treatment. Images display foci before, immediately after and 10 s after photobleaching. Representative images from three (Lsm7), six (Pab1), and five (Dcp2) independent experiments. **e** FRAP recovery curves for Lsm7-GFP, Pab1-RFP, and Dcp2-GFP foci after 2 h 2-DG stress. Since the recovery of Lsm7 foci was quick, the *x*-axis (time, seconds) is shorter than for Pab1-RFP and Dcp2-GFP. Three (Lsm7), six (Pab1), and five (Dcp2) independent experiments. **f** FRAP results depicting time (s) to full recovery after bleaching (left) and calculated half-time rate (right). Three (Lsm7), six (Pab1), and five (Dcp2) independent experiments (one-way ANOVA followed by Dunnett's T3 multiple comparisons test, with individual variances computed for each comparison). Left to right: **$p = 0.0035$, **$p = 0.0029$, *$p = 0.0134$, ns = 0.0912. Source data are provided as a Source Data file.

To further describe the in vivo physical states of SGs, PBs, and Lsm7 foci under 2-DG stress, we performed in vivo FRAP assays. Lsm7 foci have shorter half-time rate and faster full recovery after photobleaching, as compared to Pab1 granules, indicating a more dynamic nature (Fig. 7d–f). Lsm7 foci have similar half time rates as Dcp2 foci; however, the Dcp2 foci measured display a big variance (Fig. 7f). Nonetheless, the full recovery time for Lsm7 foci after bleaching is significantly shorter than for Dcp2 foci, indicating faster dynamics of Lsm7 foci (Fig. 7d–f). The overall recovery rates for Pab1 and Dcp2 in our FRAP setup are faster than what has been shown before for Pab1 (heat shock)[77], Dcp2 (log-phase)[31], and Lsm4 (PB marker, glucose depletion)[78], indicating varying condensate dynamics under different stressors (Fig. 7e).

These observations indicate that Lsm7 foci are phase-separated condensates that display more dynamic characteristics than SGs under 2-DG treatment (Figs. 5 and 7). The in vitro co-phase separation behavior of Lsm7 and Pab1 further reinforces the interconnection between the in vivo protein co-localization and the individual in vitro phase separation behavior of these proteins. The enlarging effect on Lsm7 droplet formation by RNA, as well as Lsm7's rescue of Pab1 demixing in the presence of RNA (Fig. 6), further highlights the role of Lsm7 phase-separated droplets in the initiation of Pab1 granule formation. We propose that Lsm7 may trigger Pab1 demixing and subsequent SG initiation by creating phase-separated condensates that interplay with RNA and Pab1[79,80]. Based on the presented results we identified Lsm7 as an early phase separation factor that promotes the initiation of SGs under 2-DG treatment (Fig. 8).

## Discussion

Here, we identified a set of yeast proteins co-localizing with Pab1 upon 2-DG treatment. Among these proteins, the highly conserved protein Lsm7 plays an active role in SG formation, through the formation of phase-separated condensates that further promote Pab1 nucleation. PBs have long been evidenced to promote SG formation. However, the details have not been fully elucidated. As a component of the Lsm1-7/Pat1 complex, Lsm7 has been reported to be a PB component[20]. Whether it has a function in SG formation independent from its role as a PB component hasn't been characterized before. We found that under normal growth Lsm7 does not form foci, unlike Dcp2. Furthermore, prolonged 2-DG treatment does not increase Lsm7 focus size, whereas PBs have been shown to grow in size during prolonged stress conditions[46]. In addition, FRAP assays performed on Lsm7 and Dcp2 foci show Lsm7 foci to be faster at fully recovering after

bleaching, as compared to Dcp2 foci. Unlike the other PB-linked components of the Lsm1-7/Pat1 complex, Lsm7 displays much higher co-localization with SGs under 2-DG stress. Furthermore, PBs do not seem to require Lsm7 for granule formation, but Lsm7 foci formation is clearly affected by deficient PB formation. However, in a PB deficient mutant, addition of exogenous Lsm7 could increase the number of SGs without affecting the number of PBs. Moreover, in vitro phase separation assays show that Lsm7 can form LLPS assemblies without the presence of PB components and that these condensates can co-phase separate with Pab1, and even can enhance Pab1 droplet formation at a low Pab1 concentration. Lastly, Lsm7 can protect Pab1 demixing upon addition of RNA. With these results, we, therefore, propose that Lsm7 has independent roles in SG formation, but requires assistance from PBs, in order to reach full foci formation potential.

Compared to previous studies in yeast using different stressors[4,14], our screening approach using the genome-wide yeast GFP fusion collection identified 14 strong SG-localizing proteins under 2-DG treatment. The discrepancies of the low number of SG-localizing proteins and the fact that some of the known SG core proteins are missing in the screen can be explained as follows: First, we set a very high cutoff value (60% co-localization) for isolating key SG components that are strongly co-localized with the Pab1-RFP SG marker. However, when including co-localization values below 60%, the full list of hits exceeds 100 proteins. Second, our screen is imaging-based, meaning that it is highly dependent on the images acquired from the cells. Thus, if one protein's abundance is rather low, as is the case for several known PB-associated proteins[31], it may be more difficult to detect a clear protein-GFP signal, resulting in false-negative results. Whether this can be resolved when using a higher resolution imaging system in combination with manual analysis of the images, remains to be clarified. Lastly, in our study, we used the SG marker Pab1 and 2-DG as the stress condition, which can result in differences in interaction patterns as compared to utilizing e.g., Pub1 as the SG marker[14] or $NaN_3$ as a stressor[4].

Lsm7 could also be promoting SG initiation through RNA interactions, considering that Lsm7 is an RNA-binding protein, and the recent reports propose that intermolecular RNA interactions can trigger RNP granule assemblies[71]. In agreement with this, we have shown that Lsm7 foci co-localize with RNA in vivo and require non-translating mRNA to form foci. In addition, RNA can increase Lsm7 phase separation in vitro. Interestingly, during co-phase separation of Lsm7 and Pab1, in the presence of total RNA, Lsm7 rescues the phase separation of Pab1, which

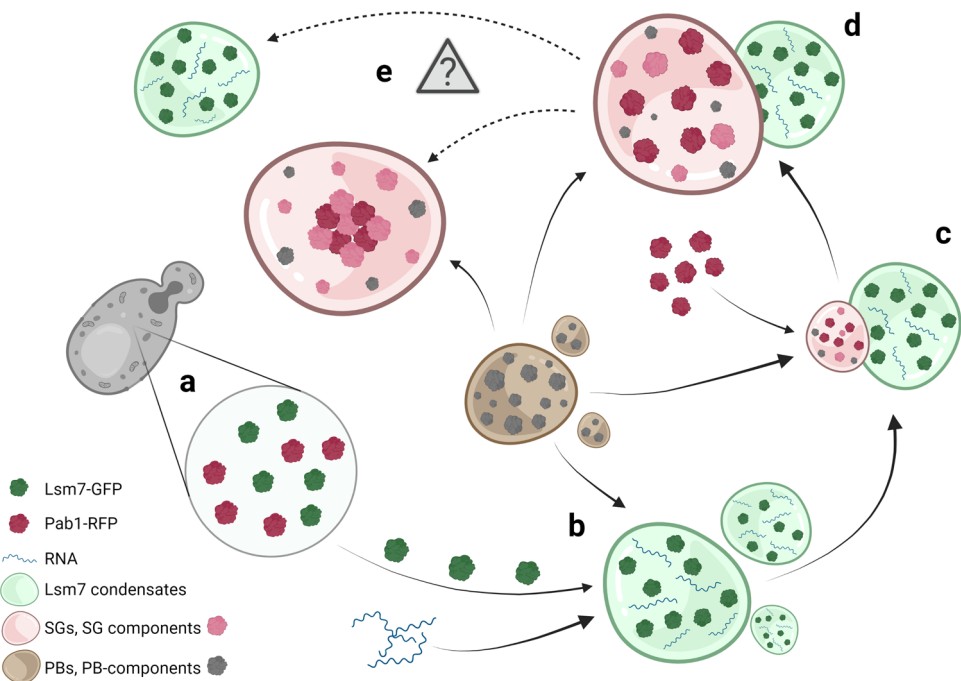

**Fig. 8 A schematic diagram showing an active function of Lsm7 phase separation in regulating SG formation under 2-DG treatment. a** Prior to stress treatment, Lsm7-GFP and Pab1-RFP are diffusely distributed in the cytosol. **b** Upon 2-DG treatment, Lsm7 quickly phase separates, creating condensates in conjunction with available RNA and supported by PB components. **c** Pab1 is recruited and starts to demix, aided by PBs and/or specific PB components. The subsequent SGs associate on the side of the Lsm7 condensates. **d** Over time SGs increase in size whereas the Lsm7 condensates remain similar in size. **e** Lsm7 condensates might not be needed for maintenance of already formed SGs, and SG core maturation. Hence, it might be that Lsm7 condensates and SGs dissociate (indicated by dotted arrows). Created with BioRender.com.

otherwise becomes reduced due to RNA addition. Accordingly, others have proposed that Pab1 needs to release RNA in order to phase separate[19]. We, therefore, propose that Lsm7 condensates further promote SG formation by rescuing Pab1 demixing in the highly concentrated RNA environment of RNP granules, most likely through substitution of Pab1–RNA interactions with enhanced Lsm7-RNA interactions and/or strong Lsm7-Pab1 condensate interactions. This does not exclude potential recruitment of other mRNPs (e.g., other Lsm1-7 complex components) by Lsm7, resulting in overall modulated intermolecular interactions and RNA availability that can further affect the phase separation process of Pab1[19].

Our super-resolution data show that Pab1 granules grow on the side of Lsm7 condensates and that Pab1 granules keep growing once formed, while the size of Lsm7 foci does not change. This further supports the notion that higher concentrations of mRNPs (e.g., Lsm7-containing condensates) could induce the initiation of Pab1 oligomerization, perhaps by functioning as a seeding scaffold[4]. However, after 120 min of 2-DG treatment, we observed more Pab1 granules than Lsm7 foci. Considering the dynamic nature of Lsm7 condensates and the fact that SGs are known to be constantly exchanging materials with the cytoplasm[15], we hypothesize that the Lsm7 droplets may "bud off" from SGs after successful SG initiation, and probably go back to a soluble state in the cytosol. This suggests that perhaps Lsm7 condensates are not required for the further maturation of SG solid cores. This notion is further supported by our findings showing that Lsm7 condensates are not needed for the maintenance of preformed Pab1 granules.

Several aspects of Lsm7 functions in SG formation remain unclear. First, although we have shown that Lsm7 foci co-localize with Pab1 granules, it needs to be further investigated whether Lsm7 and Pab1 have a direct physical interaction. Given that Pab1 has been shown to act as a physical stress sensor under heat or pH stress, and that the phase separation of Pab1 is modulated by its low-complexity domain[19], it is of importance to determine whether Lsm7 interacts with this low-complexity domain of Pab1 to influence the phase separation. Additionally, the in vivo docking phenotype and how it links to the in vitro co-phase separation phenotypes of Pab1 and Lsm7 needs to be further investigated. Moreover, the similarities and/or differences in interaction patterns and dynamics of Lsm7, observed under conditions of additional stresses, need to be clarified in more detail. Second, it is not clear what the impact would be on SG formation if the RNA-binding domain of Lsm7, and that of other SG-linked RBPs, was disrupted. Would this result in impaired SG formation? Lastly, we show that Lsm1, Lsm8, and Pat1 affect Lsm7 foci formation, at least partially through their regulation of Lsm7 cellular localization. Further studies elucidating the mechanism behind this and whether it involves the whole Lsm1-7/Pat1 complex, are necessary.

Lsm7 is a highly conserved protein. Besides its roles in RNA processing, Lsm7 has been shown to modify the toxicity of FUS[35]. Studies have also implicated Lsm7 in other disorders. For instance, multiple point mutations of Lsm7 have been found to drive tumor progression although the underlying mechanism is unknown[38]. Moreover, genome-wide RNAi screens revealed that knockdown of Lsm7 resulted in increased virus replication and infection[39,40]. This may be explained by our current data regarding Lsm7 and SG formation since SGs are known to play a role in antiviral responses[5].

Here we report a possible mechanism for regulation of SG initiation under 2-DG induced stress via Lsm7 phase-separated condensate formation. This process might represent a mechanism for the initiation of SG formation when energy and nutrient supply is limiting, a condition under which most of the microbial biomass in the world is believed to exist[81]. Other components and signaling pathways regulating the SG formation under these

conditions or other types of stress remain to be elucidated. Further studies are also required to elucidate the role of Lsm7 in SG formation in mammalian cells, especially when it comes to the mechanisms underlying SG-induced drug resistance or the relationship between SGs and age-related diseases such as cancer and neurodegeneration.

## Methods

**Experimental design**. This study aimed to elucidate the mechanisms behind SG formation under 2-DG stress in yeast. To this end, Pab1-co-localizing proteins were identified through an automated imaging-based phenomic screen based on the SGA method[28]. Promising candidates from this screen were further analyzed, and one such candidate was selected to elucidate its role in SG formation and overall condensate properties.

**Yeast cell culture**. All the strains used in this study were from the BY4741/4742 background (strains are listed in Supplementary Data 1). Strains were grown at 30 °C or at the indicated temperatures. The temperature-sensitive (TS) allele *dcp2-7* strain[82] was precultured at RT and switched to 37 °C when performing experiment. Yeast-rich medium (YP) containing 1% Bacto yeast extract and 2% Bacto peptone was supplemented with 2% glucose (YPD). Yeast minimal medium (YNBD) contained 0.67% Difco yeast nitrogen base without amino acids and 2% glucose. Supplements essential for auxotrophic strains were added to 20 mg/l for bases and amino acids (complete). Leucine (SC-Leu), histidine (SC-His), or uracil (SC-Ura) was omitted when appropriate.

**Yeast strain construction**. The null mutants were constructed by PCR amplification by insertion of selective markers including *LEU2*, *natMX4*, and *kanMX4* (primers are listed in Supplementary Data 2). The strains used for protein expression and localization analyses were picked directly from the yeast GFP collection or were constructed by using standard PCR to either integrate an *Aequorea victoria* GFP (S65T) or enhanced GFP[83], mRuby2[84], or RFP[85] tag into the yeast chromosome (C-terminal of ORF) through homologous recombination and the constructs were expressed using endogenous promoters[28]. The Lsm7-5xFlag strain was constructed by tagging 5xFlag onto the C-terminal of the Lsm7 protein. PCR primers for 5xFlag (*LEU2* marker) amplification are listed in (Supplementary Data 2). The Pab1-RFP *lsm4ΔC* strain was constructed based on BY4741 Pab1-RFP, by the partial deletion of the C-terminal 97 amino acids of the Lsm4 with *LEU2* insertion[42].

Based on the IDR prediction by IUPred2 (https://iupred2a.elte.hu/), the first and most prominent IDR was selected for strain construction. For construction of the IDR truncated version of *LSM7-GFP*(HIS) (*lsm7ΔIDR*), one PCR was performed to amplify *lsm7Δ1-22*, the other one to amplify GFP (remove ATG from GFP) followed by a bridge PCR using an equal mix of the two PCR products as the template (Supplementary Data 2). For construction of the truncated versions of *LSM7-GFP* (*lsm7Δ39-53* and *lsm7Δ90-103*), the hydrophobic regions of Lsm7 were commonly predicted by three different online tools; structural predictions and sequence propensities (hydrophobicity clusters) using FELLS (http://old.protein.bio.unipd.it/fells/entry/LBD0KzUVuBCAGeCVPV5bSS8Hodg?name=Lsm7&session=5f5b7d1832d0d67607ccd8de), prediction of "hot spots" of aggregation in polypeptides using AGGRESCAN (http://bioinf.uab.es/aggrescan/), and hydrophobicity prediction by ExPASy-ProtScale (Kyte & Doolittle, https://web.expasy.org/protscale/). Subsequently, the residues 39–53 and 90–103 were found to be hydrophobic regions and were selected. Construction of *lsm7Δ39-53* required three PCR steps. First, two PCR reactions to amplify 1–39 and 53 to end fragments, followed by a bridge PCR to get the full *lsm7Δ39-53*, and lastly another bridge PCR with GFP to get *lsm7Δ39-53-GFP*. Same strategy was used for *lsm7Δ90-103-GFP* construction (Supplementary Data 2). All three *LSM7* constructs were cloned into plasmid pYM28 between *Hind*III and *Bam*HI and sequenced to confirm the deletion and proper coding. The upstream and downstream sequences of *LSM7* were used as the homology sequence for transformation (homologous recombination) into BY4741 Pab1-RFP strain. Final truncation mutants were selected on SD minus HIS plates.

For construction of the Lsm7-Spacer-eGFP strain, the spacer sequence was amplified and flanked by *Sal*I and *Asc*I sites from a plasmid bearing the spacer (p416GAL1-GFP-spacer)[68]. This fragment was then inserted upstream of *LEU2* into a plasmid and a set of primers (Supplementary Data 2) was subsequently designed to amplify the Spacer-*LEU2* region. The forward primer contained part of eGFP ORF (55 bp upstream of the stop codon) and part of the spacer (19 bp from the start site), while the reverse primer consisted of part of *LEU2* (21 bp upstream of the stop codon) and 55 bp of *LSM7* downstream of the stop codon.

**Genome-wide high content screening for SG components**. To incorporate the SG marker (Pab1-RFP) into the yeast GFP collection[28], Pab1-RFP was introduced into the yeast query strain Y7039. The SGA method was then applied[86]. To screen the SG components in this collection, the cells were precultured in SGA final medium (SD-Leu/Arg/Lys/His+S-AEC, Canavanine, and Hygromycin B) at 30 °C for 2 days. The precultured cells were then diluted to OD_600 0.05 in SD minus His

medium and grown with shaking at 30 °C. When the OD_600 of most wells had reached 0.8 (about 10 h), 2-DG was added to a final concentration of 400 mM for 2 h with continuous shaking. The cells were then fixed with 3.7% formaldehyde for 30 min and washed twice with PBS. Imaging was performed with an automated cellular imaging system (ImageXpress MICRO (MDC)). After image acquisition, the images and any co-localization were analyzed manually using the composition function of the MetaXpress (Version 3.1) software. The hits from the screen were then manually confirmed by using a conventional microscope (Zeiss AxioObserver.Z1, Germany). The degree of co-localization with Pab1 was calculated by dividing the number of foci containing both Pab1-RFP and the GFP-tagged protein with the number of total Pab1-RFP granules. The opposite calculation, number of co-localizing foci divided by total number of GFP foci, is also included in Supplementary Data 3.

**Interaction network analysis**. For the SG component screen, the interaction network diagram of the hits was extracted from the interaction analysis using Osprey 1.2.0, and the physical interactions between confirmed hits were added according to the BioGRID interaction database (https://thebiogrid.org/).

**Bioinformatic analysis**. Intrinsically disordered regions and disordered binding regions were predicated by IUPred2 (https://iupred2a.elte.hu/) default program[87]. The prediction of prion-like domains was done with PLAAC (http://plaac.wi.mit.edu/) by using default settings[88]. The prediction of LLPS propensity in yeast was obtained by using catGRANULE (http://www.tartaglialab.com/)[63–65]. Hydrophobicity prediction was made by ExPASy - ProtScale (Kyte & Doolittle, https://web.expasy.org/protscale/) with default settings. Yeast Lsm7 and human Lsm7 protein sequence alignment was done by using CLUSTALO program (1.2.2).

**Stress conditions**. For all stress conditions, cells were grown to OD_600 of 0.5–0.6 in YNBD complete media and then exposed to the indicated stress conditions: addition of 400 mM 2-DG for 2 h; glucose depletion (wash and resuspension in media without glucose) for 120 min; 44 °C for 15 min; 1% NaN_3 for 30 min. For stationary phase, cells were grown until OD_600 > 4. Samples were collected, fixed, and imaged as described below.

**Digitonin and 1,6-hexanediol treatments**. To study the properties of Lsm7 foci and Pab1 granules induced by 2-DG, log-phase cells were treated with 400 mM 2-DG for 2 h, followed by treatments with 10 μg/ml digitonin or 10 μg/ml digitonin plus 10% 1,6-hexanediol for another hour. Digitonin was used to make the yeast cells more permeable to 1,6-hexanediol. Cells were then fixed with 3.7% formaldehyde and washed twice with PBS. To further study the 2-DG-induced formation of Lsm7 foci and Pab1 granules with or without 1,6-hexanediol, log-phase cells were pretreated with 10 μg/ml digitonin or 10 μg/ml digitonin plus 10% 1,6-hexanediol for 30 min, followed by 400 mM 2-DG treatment for 2 h. For the washout assay, the digitonin and/or 1,6-hexanediol were then washed out from the media and cells were treated with 2-DG for another 2 h. Samples were taken at indicated time-points, fixed and imaged as described below.

**Cycloheximide treatment**. Cells were grown to an OD_600 of 0.5. To visualize the inhibiting effects of cycloheximide on Lsm7 foci formation, cycloheximide was added to a final concentration of 100 μg/ml for 30 min followed by addition of 400 mM 2-DG for 2 h. For the Lsm7 foci disassembly assay, cells were first stressed with 400 mM 2-DG for 2 h, followed by addition of 100 μg/ml cycloheximide for an additional 2 h. For both setups, water was added instead of cycloheximide to the control. Samples were collected, fixed, and imaged as described below.

**Puromycin treatment**. A puromycin-sensitized triple mutant (*pdr1Δ pdr3Δ snq2Δ*) was used for this assay. The cells were grown to an OD_600 of 0.5, followed by treatment with both 400 mM 2-DG and 1 mM puromycin for 1 h. Water was added instead of puromycin to the control. Samples were collected, fixed, and imaged as described below.

**Fluorescence microscopy**. Cells were grown to an OD_600 of 0.5 with or without treatments as described above and fixed with 3.7% formaldehyde for 30 min, followed by two times washing with PBS. For DAPI staining, samples were pretreated with EtOH, washed, and resuspended in 1 μg/ml DAPI solution before imaging. A Zeiss Axiovert 200 M fluorescence microscope (100 × 1.4 NA oil objective) was used to obtain images using GFP, RFP, and DAPI channels.

**FRAP**. FRAP of Lsm7–GFP, Dcp2-GFP, and Pab1-RFP foci (2 h 2-DG) was performed with a Zeiss LSM880 Airyscan confocal microscope. The cover slips were coated with 0.25 mg/mL concanavalin A to immobilize the cells. Using a 63×/1.4 oil objective, the regions of Lsm7–GFP and Dcp2-GFP foci were bleached using a laser intensity of 90% at 488 nm and for Pab1–RFP foci using a laser intensity of 90% at 561 nm. The recovery time was recorded for the indicated times. Analysis of the recovery curves was carried out with ZEN 2.3 and GraphPad Prism 9.

**Fluorescent signal intensity analysis**. The fluorescent signal intensity for Dcp2-GFP was measured by using the software ImageJ (1.53c) (integrated density). The value was presented as relative density to that of BY4741 with 2-DG.

**Expression and purification of proteins**. To express Lsm7-GFP protein, the Lsm7-GFP sequence from the yeast-GFP collection was cloned into a pET28a-vector via the $NdeI$ and $XhoI$ restriction sites using the standard cloning methods. Lsm7-GFP mutants and Pab1-RFP sequence were amplified from corresponding yeast strains, then cloned into pET32a-vector for Lsm7ΔIDR-GFP and pET28a-vector for Lsm7Δ90-103-GFP as well as Pab1-RFP, by using GeneArt™ seamless cloning and assembly kit (Thermo Fisher Scientific). Recombinant 6xHis-tagged Lsm7-GFP, Lsm7-GFP mutants, and Pab1-RFP constructs were overexpressed in *E. coli* Rosetta 2 (DE3) pLysS. The strains were grown in LB medium containing 50 μg/ml kanamycin at 37 °C until $OD_{600}$ reached 0.7–0.8, followed by induction by 0.5 mM IPTG, at 23 °C overnight. Cells were harvested by centrifugation (10 min, 6370 g, JA-10, Beckman) and lysed via sonication on ice, in buffer containing 50 mM $NaH_2PO_4$, 0.5 M NaCl, 10 mM imidazole, pH 8.0, and 1 mM PMSF. Cellular lysates were clarified by centrifugation at 35,270 g (JA-17, Beckman) for 1 h at 4 °C. The supernatants were loaded onto a 20 ml gravity chromatography column containing 8 ml $Ni^{2+}$-NTA resin (QIAGEN). After washing 2.5 resin volumes respectively with washing buffers containing 50 mM $NaH_2PO_4$ and 0.5 M NaCl with increasing imidazole concentrations (10 mM, 25 mM, and 50 mM), the target proteins were eluted with buffer (50 mM $NaH_2PO_4$ and 0.5 M NaCl, 250 mM imidazole, pH 8.0). The eluted proteins were loaded into dialysis tubing with thrombin and dialyzed with 1× PBS buffer (137 mM NaCl, 2.7 mM KCl, 10 mM $Na_2HPO_4$, 1.8 mM $KH_2PO_4$, pH 7.4) overnight, for removal of the N-terminal His tag. The proteins were subsequently loaded onto the $Ni^{2+}$-NTA column again and eluted with buffer (50 mM $NaH_2PO4$ and 0.5 M NaCl, 10 mM imidazole, pH 8.0) to remove tags and uncleaved protein. The proteins were concentrated by using Vivaspin® 20 (Sartorius, 10 kDa MWCO) centrifugal filters at 3,000 × g and loaded onto an Äkta Pure system (GE Healthcare) equipped with a Superdex 200 10/300 GL column (GE Healthcare) for size exclusion chromatography (SEC) and eluted with the storage buffer (50 mM HEPES, pH 7.4, 400 mM NaCl). For purification of Pab1-RFP protein, the eluted protein from the $Ni^{2+}$-NTA column was exchanged to buffer containing 50 mM TrisHCl, pH 7.4, 50 mM NaCl, and loaded onto a 5 mL HiTrap heparin HP column, and eluted over a NaCl gradient. The sample was concentrated and purified by a Superdex 200 10/300 GL column (GE Healthcare) and eluted in storage buffer (50 mM HEPES, pH 7.4, 400 mM NaCl). After SEC, the proteins were concentrated again by using Vivaspin® 20 centrifugal filters at 3000 × g. The purity of the proteins was assessed by SDS PAGE. Protein concentration was determined at 280 nm with the Nanodrop spectrophotometer (Thermo Fisher Scientific), using the corresponding theoretical extinction coefficient for each protein. The protein was flash-frozen in the protein storage buffer using liquid nitrogen and stored at −80 °C.

**Fluorescence microscopy for phase separation assays**. Purified fluorescently labeled proteins were imaged using 100 × 1.4 NA oil objective, on a wide-field fluorescence microscope Zeiss Observer Z1. Reactions for phase separation assay were prepared in tubes and transferred to slides for imaging. Lsm7 droplet formation was induced at room temperature by adjusting the salt concentration to 150 mM NaCl and adding crowding agent Dextran 70 or Ficoll 400, keeping the concentrations of other buffer components the same (pH 7.4). The condensate sensitivity to changes in pH (pH 7.4, pH 6.5, or pH 5.5) and salt concentration (150 mM, 500 mM, or 800 mM) was studied by adjusting the components accordingly. Droplets were allowed to grow for 30 min before imaging. For phase separation assays in which 200 ng/μl total yeast RNA (Thermo Fisher Scientific) was added, all images were captured after 5 min of phase separation induction.

**Super-resolution three-dimensional structured illumination microscopy**. Cells carrying both Pab1-RFP and Lsm7-eGFP were treated and fixed with 3.7% formaldehyde and washed twice with PBS (pH 7.4). For 3D-SIM, the ELYRA PS.1 LSM780 setup from Zeiss (Carl Zeiss, Jena Germany) was used[89]. 3D-SIM images of the protein foci (Lsm7-eGFP) and SGs (Pab1-RFP) were taken with a 100×/1.46 plan-apochromat oil-immersion objective with excitation light wavelengths of 488 nm and 561 nm, respectively. Z-stacks with an interval of 100 nm were used to scan the whole yeast in 3D-SIM. For acquisition and super-resolution processing and calculation as well as for 3D reconstruction, the Zen2012 software (Carl Zeiss, Jena Germany) and Imaris 7.2.3 were used. The ELYRA System was corrected for chromatic aberration in *x*-, *y*-, and *z*-directions using multicolor beads, and all obtained images were examined and aligned accordingly. The sizes of SGs and Lsm7-eGFP foci were quantified by measuring the average areas (converted from the quantified pixels) of the corresponding signals and then calculating the diameters. The association of Lsm7-eGFP with SGs (Pab1-RFP) under 2-DG treatment for 120 min was demonstrated by constructing a 3D-Surface using the Imaris 7.2.3 software.

**Western blotting assay**. The total yeast protein extraction method[90] was used with modification. About 1.0 unit $OD_{600}$ of yeast cells were harvested and incubated in 1 ml of 0.2 M NaOH for 20 min on ice. It was then resuspended in 50 μl of HU sample buffer (8 M urea, 0.2 M Tris–HCl, pH 6.8, 1 mM EDTA, 5% SDS, 1% β-mercaptoethanol, 0.0025% bromophenol blue) and heated for 10 min at 70 °C. All samples were electrophoresed on 10% Tris-HCl/SDS-polyacrylamide gels (BioRad) and transferred onto polyvinylidene difluoride membranes (Merck Millipore). The membranes were then exposed to primary antibodies; mouse anti-Pgk1 (1:1000, Invitrogen, Cat# 459250, monoclonal (22C5D8)), mouse anti-GFP (1:200, Santa Cruz, Cat# sc-9996, monoclonal (B-2)), rabbit anti-GFP (1:5000, Abcam, Cat# ab6556; 1:10000, Abcam, Cat# ab290) or rabbit anti-RFP (1:2000, Abcam, Cat# ab62341), followed by secondary antibodies; HRP conjugated goat anti-mouse IgG (H + L) (1:5000, Figs. 1e and 2c), 1:3000 (Supplementary Figs. 2c and 4b), Invitrogen, Cat# 62-6520) or goat anti-rabbit DyLight 650 (1:5000, Invitrogen, Cat# 84546). Relative protein expression was analyzed by the Odyssey® imaging system (Licor) or Bio-Rad ChemiDoc MP Imaging System and was normalized to Pgk1. For an example of presentation of full scan blots, see the Source Data file.

**SYTO RNASelect green fluorescent cell stain**. Cells were grown to an $OD_{600}$ of 0.5 followed by 2 h of 2-DG treatment. After treatment, cells were pelleted and resuspended in PBS buffer with 500 nM SYTO RNASelect green and incubated at 30 °C for 15 min. When labeling was completed, the buffer was removed and the cells were rinsed with PBS. The cells were then imaged under fluorescence microscope (Zeiss Axiovert 200 M) using FITC and RFP channels.

**In situ proximity ligation assay**. The proximity ligation assay (PLA) was performed to determine protein interaction as reported before[91,92]. To determine interaction between Lsm7 and Pab1, BY4741 strain and Lsm7-5xFLAG were cultured to $OD_{600} = 0.5$, and treated with or without 400 mM 2-DG for an additional 2 h. Cells were fixed with 4% PFA for 20 min and washed with 0.1 M potassium phosphate pH 7.4. The cell wall was digested by Zymolyase according to the manufacturer's protocol. Briefly, the digested cells were placed on the poly-lysine coated slides and immersed in methanol and acetone in turn. After blocking with 3% BSA (in PBS), the cells were incubated with primary antibodies against Pab1 (1:100, EnCor Biotechnology, Cat# MCA-1G1, monoclonal (1G1)) and Flag (1:100, Sigma-Aldrich, Cat# F7425) overnight at 4 °C. After washing, the slides were incubated with PLA probes (Duolink® In Situ Red Starter Kit, Cat# DUO92101, Sigma-Aldrich) at 37 °C for 1 h. Ligation and DNA amplification were then carried out at 37 °C for 30 min and 100 min, respectively. Images were taken by a conventional microscope (Zeiss AxioObserver.Z1, Germany).

**Statistical analysis**. No statistical methods were used to predetermine sample size. Appropriate statistical analyses were performed dependent on the comparisons made as described in the text and figure legends. One-way or two-way ANOVA following Dunnett's, Tukey's or Šídák's test, or two-tailed unpaired *t* tests were performed using GraphPad Prism version 9 (Graphpad, Inc.). *P* values are designated as *$P < 0.05$, **$P < 0.01$, ***$P < 0.001$ and ****$P < 0.0001$. All graphs show mean and error bars representing standard error of the deviation (S.D.)

**Reporting summary**. Further information on research design is available in the Nature Research Reporting Summary linked to this article.

## Data availability

The authors declare that all data supporting the findings of this study are available within the paper and/or the Supplementary Information/Source data file. Protein domain predictions and interaction analysis were performed by using these openly available databases: IUPred2, PLAAC, catGRANULE, ExPASy – ProtScale, FELLS, AGGRESCAN and BioGRID. For further details see Methods section. Source data are provided with this paper.

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

## Acknowledgements

The authors would like to thank Charles Boone for the generous contribution of the Moby plasmids, and Paola Branduardi for kindly sharing strains. We also thank the Centre for Cellular Imaging at University of Gothenburg for microscopy support, as well as Simon Alberti and Marc Pilon for their careful and critical reading of this manuscript. This work was supported by grants from the Swedish Cancer Society (CAN 2012/601, CAN 2015/406 and CAN 2017/643 and 19/0069 to B.L.; 19/0133 to P.S.), the Swedish Natural Research Council (VR 2011-5923, VR 2015-04984 and VR 2019-03604 to B.L.), and the Carl Trygger Foundation (CTS 14: 295, to B.L.). The research leading to these results has received funding from the People Program (Marie Curie Actions) of the European Union's Seventh Framework Programme (FP7/2007-2013) under REA grant agreement n°608743. J.Z. received a fellowship from CSC and from KU Leuven (grant C14/17/063).

## Author contributions

B.L. conceived the project and B.L., L.C., C.M.G., B.M.B., J.W., H.L., and P.S. supervised the research. M.L, L.C., D.Z., S.J., J.Z., X.H., X.Y., A.K., L.C.M, J.T., Y.G., D.Y.Z., and X.Z performed the experiments. M.L., L.C., and B.L. analyzed the data and wrote the manuscript, which was further edited by the other co-authors.

## Funding

## Competing interests

The authors declare no competing interests.
