## [Peer Review File · Nature Communications]

Lsm7 phase-separated condensates trigger stress granule formationREVIEWER COMMENTS

Reviewer #1 (Remarks to the Author):

In this manuscript the authors conclude that Lsm7 condensates play a key role in initiating SG assembly based on a series of observations:

- Upon 2DG treatment, epitope tagged Lsm7 foci and SGs visualized using epitope tagged Pab1 colocalize as seen under the microscope and verified by PLA assay.
- During 2DG treatment, SG formation monitored using epitope tagged Pab1 and Pbp1 is impaired in lsm7 Δ background although PB formation was unaffected in lsm7 Δ background suggesting that role of Lsm7 in facilitating SG formation is not related to PB formation.
- Lsm7 overexpression enhanced SG formation.
- Internal deletions removing the aggregation prone domains in Lsm7 impair Lsm7 foci formation and SG formation. Similarly, fusion of spacer sequence from Sup35 to the C-terminus of Lsm7 decreased the formation of both Lsm7 foci and SGs.
- Time course study of Lsm7 and SG formation during 2DG treatment revealed that Lsm7 foci form well before SGs form and SGs stay bound to the side of Lsm7 foci.
- Colocalization data also revealed that Lsm7 foci contain RNA.
- In vitro, purified Lsm7 was able to undergo LLPS.
- Pre-treatment of cells with 1,6-hexanediol prevented SG and Lsm7 foci formation. However, treatment with 1,6-hexanediol after the Lsm7 foci and SGs are induced by 2DG, resulted in loss of only Lsm7 foci suggesting that Lsm7 foci are not needed to maintain SGs.

The observations are very interesting and overall, this is a very well carried out study. However, the manuscript can be strengthened by addressing some important questions especially about the Lsm7 foci which the authors argue is involved in initiating the Pab1 containing SGs.

The authors show that the Lsm7 foci contain RNA. However, what about other Lsms? Do the Lsm7 foci contain free Lsm7 or the entire Lsm1-7 complex? Importantly, is the ability of Lsm7 to get incorporated into the Lsm1-7 complex critical for foci formation and triggering SG formation? For example, if point mutations impairing the incorporation of Lsm7 into the Lsm1-7 complex are introduced into Lsm7, do Lsm7 foci and SGs still form under those conditions? The authors show that knocking out PB components like Dcp2 impairs Lsm7 foci formation. Does this also happen in lsm1 Δ background? If yes, under those conditions, do the fewer Lsm7 foci present still support SG formation? It is known that over

expression of Lsm8 can sequester Lsm2 through Lsm7 proteins in the nucleus. Does that impair Lsm7 foci formation and SG formation? Similarly, do point mutations in the conserved RNA binding loops of Lsm7 impair Lsm7 foci formation and SG formation? Addressing some of these questions will provide important insight into the nature of Lsm7 foci (and the mechanisms by which they facilitate SG formation) and make the story more complete.

The in vitro experiments are very interesting. However additional data to support their relevance to in vivo observations would help. For example, do the internal deletions in Lsm7 (that remove the aggregation prone domains) or fusion of spacer sequence to Lsm7 impair LLSP of Lsm7 in vitro also?

Other questions/comments:

Fig1: What is the impact of 2DG treatment on the expression of epitope tagged Lsm7?

Fig2D: Western should also include Pbp1-GFP control

Fig2E: How is PB formation induced and what do the time points (20 min and 120 min) refer to?

Reviewer #2 (Remarks to the Author):

In this paper the authors describe a facilitating role for Lsm7 in the formation of 2-deoxyglucose induced stress granules in yeast, based on results from a prior published GFP library co-localization screen. They show during a 2DG stress timecourse that Lsm7 forms foci (aka seeds, scaffolds) prior to Pab1 following 2DG stress; Pab1 foci physically dock on the side of Lsm foci continue to enlarge whereas Lsm7 retain a certain size. Lsm7 is shown to phase separate in vitro, through these experiments do not really provide significant insight. A distinct sensitivity of Lsm7 and Pab1 foci to 1,6 hexanediol is observed (Lsm7 foci are sensitive, Pab1 foci are not), which reportedly affects only liquid-like assemblies, not solid assemblies – though it should be noted there are contradictory findings in the field about the use of this chemical, and how it should be interpreted.

While there are minor fixable issues with the data as presented, and some overstatements, the paper is relatively short and lacking in novelty, impact or mechanistic understanding that would warrant publication in a high impact journal. I have suggested new experiments, but the authors are probably best advised submitting elsewhere.

Major points:

1.) Clearly many granule assembly promoting proteins have been reported, and many identified that phase separate. Also, step-wise assembly of granule assembly has been reported in yeast and human models (e.g., Wheeler eLIFE 2016; Cirillo et al, Current Biology 2020). So there isn't a great deal of novelty here as of yet. A stronger justification as to why Lsm7 was chosen for follow up analysis in the screen would also help, since as a known PB protein, the degree of new insight one could gain already seems a potentially limited.

2.) As the authors are surely aware, in yeast, SGs and PBs spatially overlap under weak stress conditions, or in the early stages of some stress events. I rather suspect 2-DG, given the low number of foci/cell, and cells with foci, would show strong co-localization of Pab1 with Dcp2 or other PB markers. Have they ever checked this? If so, it would be useful to know if Lsm7 (being better known as a PB protein before this given it's function in the Pat1/Lsm complex), shows any foci and co-localization with PB markers in the absence of stress. i.e., add a no stress control to Figure 1D. This would change the perspective on Lsm7 uniquely seeding SGs, to the idea that this PB-localized protein seeds SGs (similar to what was suggested for PBs aiding -Glu SG assembly in Buchan et al, 2008).

- Related to this point, Lsm7 is part of the Lsm complex. The authors should check other Lsm nulls to see if the same effects are seen or not. If not, then it at least suggests the possibility of effects beyond function in the Lsm complex. Note; Lsm4 is already implicated in PB assembly, and so this at least seems an obvious related factor to check for Pab1 effects in their 2DG system.

Minor points:

1.) As the authors acknowledge in the discussion, the observation of only 14 proteins in SGs is probably an artefact of their unusually high cut-off (60% GFP foci-co-localizing with Pab1 RFP). Since SG composition in yeast has something in the ballpark of >100 verified proteins (by candidate microscopy, and validation from SG purification studies e.g., Jain et al, 2016), it undermines confidence in the screen to not see so many known SG proteins in their list. What happens if you reduce the bar to something much smaller? This data could be included as a supplement; knowing the degree of localization of protein X in a SG below 60% would be informative.

2.) More detail on microscopy co-localization approach would be appreciated (there is mention of the software used only; in Fig1 legend, noted that co-localization was done manually – does that apply to the whole original screen, or just follow up testing?). Also, it is possible that some GFP-tagged proteins may be present in SG-distinct foci, implying other roles. Thus, it would be helpful if the authors did the

opposite calculation to what they've done in 1B (i.e., divide number of GFP foci with RFP co-localization by total GFP foci).

3.) Fig S2: Based on previous Parker lab data, in which inactivating decapping enzymes (or Xrn1) leads to strong increases in PBs, whereas Edc3 inactivation reduces PBs (presumably due to loss of scaffolding functions), it's slightly surprising to see Lsm7 foci decrease in both cases. Did they check PB phenotypes in these cell lines? Could they do a co-localization with a PB marker (e.g. Dcp1)? If Pb phenotypes in these nulls are as mentioned above, it would be further evidence that Lsm7 effects on SG formation are independent of PB effects (which is a point they make in Fig2).

4.) The authors introduced 3 terms in the paper regarding domain signatures of Lsm7; aggregation prone domains (Fig 3A-B – they make truncations of these), IDRs and PLDs. They should make it clearer how (if at all) these overlap, and why they prioritize aggregation domains for analysis.

- Having shown that perturbation of Lsm7 affects SG foci, it would be prudent to check 1-2 other SG markers (e.g. Pbp1, Pub1, Ded1, eIF4G etc) to be sure that Lsm7 effects are not limited to only Pab1. I realize they do this for the null in 2B, but it's relatively little extra work.

5.) Figure 3C – the rationale to the C-terminal spacer tag experiment isn't exactly clear; i.e. why did the authors think this would impair Lsm7 function similar to a KO or truncation? Is the thought that having a charged sequence added may repel Lsm7 from interacting with itself.

6.) 3E: Why not show data from the 4 biological replicates averaged?

7.) 3H should be quantified in terms of co-localization. Also might consider to oligo-dT instead of SYTO if specifically interested in mRNA.

8.) The authors mention that the composition of Lsm foci is unclear. That could be assessed either unbiasedly (with Lsm7 purification coupled to prior centrifugation to select for higher molecular weight complexes + mass spec; akin to the SG purification steps in Jain et al), or with candidate approaches using reasonable guesses (e.g., other Lsm components, Pat1 etc). Making mutants in such components if they're established to be in the same foci, and assessing SG/PB effects would then give a little more insight into how SGs are being seeded. Note: The Lsm4 delta c mutant data in Buchan et al 2008 isn't terribly different in terms of magnitude of effect compared to what is seen here; SG reduced about 2 fold, whereas PBs largely unaffected.

9.) Respectfully, these types of single protein focused phase separation experiments don't reveal much in my view. Numerous proteins with IDRs, PLDs etc undergo phase separation at a suitably high concentration, with enough crowding agent, and basically all are impeded by high salt. It would be slightly more informative if other variables were introduced to this experiment such as:

- Lsm4 mutations (e.g. from 3B?) to establish which domains are key for phase separation properties.
- Presence/absence of RNA to establish if this influences ability to phase transition
- Presence/absence of other SG proteins of interest. Can you recapitulate Pab1-Lsm7 docking phenomenon from 3F? Do you need the rest of the Lsm complex? Etc.

10.) Hexanediol is indeed used by a number of labs (Alberti and others) to make assertions about the liquid state of condensates. I suggest the authors at least consider (even better acknowledge) Wheeler et al, eLIFE 2016, which shows that hexanediol stresses yeast in numerous ways, and induces SGs in yeast. Why do the authors think they see something different than the Parker lab here? If they in fact also see SG induction in response to hexanediol, might this complicate their data in Figure 5? Perhaps it would be worth checking if SGs are still impaired in assembly in an Lsm7 foci KO.

Other additional experimental suggestions:

- a.) FRAP analysis of Lsm7 foci, in comparison to PB or SG markers (in vivo, maybe in vitro). Are there significantly different dynamics explaining distinct assembly/disassembly behaviors?
- b.) Are Lsm7 foci dependent on translation repression? Do they still form if cells are previously treated with CHX prior to 2-DG treatment? Is their disassembly facilitated by CHX treatment? What about puromycin (same idea, opposite outcomes expected).
- c.) Extend analyses to mammalian cells, in an interesting cell and stress context model.
- d.) Personally, I would be interested more as to why a factor like Iki3 (linked to RNA tRNA modification and RNAP II function) is in SGs. In any case, doing at least a null analysis (or over-expression?) and Pab1-RFP microscopy for all the hits previously not examined would give potentially yield greater insight as to undiscovered regulators of SG assembly. SG disassembly phenotypes could also be examined potentially with something like degron alleles.

General narrative comments:

A few statements are overblown in the paper. Use of words like essential (e.g. line 257) and required (line 31, 86, 221, to described SG assembly phenotypes that while significant, are modest in fold change. These should be adjusted.

Also, the authors claim in their abstract to identify Pab1-association proteins in their abstract (which I would take to mean physical interaction), and that Pab1 directly interacts with Lsm7. In fact, the only assay presented regarding Lsm7-Pab1 interaction is a proximity ligation assay (Fig 1E), which isn't direct proof of interaction (just close proximity). These claims should be adjusted, or experiments conducted (e.g., recombinant protein IPs, ideally with mutant Lsm7 constructs they went to the trouble of making) to show whether these proteins bind, and via what domains. Otherwise, even a standard IP would add something.

Reviewer #3 (Remarks to the Author):

In this manuscript, the authors identify Lsm7 as a key early component of 2-deoxyglucose-induced stress granules. Although many components and regulators of stress granules have been identified, Lsm7's early (preceding Pab1 condensation) and almost-essential role in stress granule nucleation make this finding particularly interesting. Overall, the experiments are well done and clearly described. The authors nicely establish that although Lsm7 is important for stress granule formation, it is dispensable for the ongoing stability of these assemblies. Nevertheless, there are a few issues that should be addressed.

1. It would strengthen the manuscript to test whether Lsm7 plays a role in SGs induced by other stresses. Either result would be interesting, but such experiments would provide insight into Lsm7's mechanism of action, revealing whether its assembly is a specific response to 2-deoxyglucose-induced stress, or a more generic stress response.
2. The authors consistently quantify SG and P-body formation based on the fraction of cells with foci. However, this would seem to provide an incomplete picture, as many of the mutations seem to affect not only the number of foci, but also the intensity of these foci. Thus, it seems important to quantify both the number of cells with foci, and the fraction of signal in these foci. This is particularly important in cases where the authors are arguing that a mutation does not impact assembly. For example, the authors argue that Lsm7 deletion does not reduce P bodies; however, by eye it appears that although most cells still have P bodies, they are on average less intense.
3. For Lsm7 overexpression (Fig 2C), it is important to show cells both before and after stress, to ensure that overexpression isn't inducing assembly independent of stress.

4. In Figure 3, the authors should confirm that their mutations (deletion of aggregation-prone segments and addition of the M domain) don't impact expression, as they have already shown that Lsm7 expression can influence SG formation.

5. In Figure 3, the authors fuse the Sup35 M domain to their construct to prevent assembly. While this indeed worked, the authors give little rationale for choosing this domain. The M domain is a regulator of stress-induced assembly (Franzmann et al, 2018). While it prevents formation of irreversible aggregates by Sup35, it promotes formation of gels in response to stress. Given this complex relationship with protein assembly, it is a surprising choice for a solubilization domain, so a more thorough explanation for this choice would be helpful.

6. The authors propose multiple models by which Lsm7 could initiate stress granule formation, including trapping RNA that facilitates Pab1 binding, or directly interaction with Pab1. It is therefore surprising that the authors don't test include Pab1 and RNA in their in vitro experiments, to see if Lsm7 promotes Pab1 assembly, and whether RNA affect either Lsm7 assembly or Pab1/Lsm7 co-assembly.

7. The authors repeatedly describe Lsm7 as "essential" for 2-deoxyglucose-induced SG formation (including in the abstract), yet according to their data (Fig 2), Lsm7 deletion reduces, but does not eliminate SG formation. Thus, while it is clearly important, it is not accurate to say that Lsm7 is "essential."

We thank all the reviewers for their insightful comments and suggestions that have helped us in making our manuscript stronger. As a result of addressing all the comments from the reviewers, we have added or modified a total of 34 sub-figures and included one supplementary table with new data. Please see below for our point-by-point response to the comments. We would like to mention that some of the experiments required repeating due to the inclusion of additional strains of interest and some values in the previous figures (e.g. foci formation and co-localization percentages) have been updated with new values.

REVIEWER COMMENTS

Reviewer #1 (Remarks to the Author):

In this manuscript the authors conclude that Lsm7 condensates play a key role in initiating SG assembly based on a series of observations:

- Upon 2DG treatment, epitope tagged Lsm7 foci and SGs visualized using epitope tagged Pab1 colocalize as seen under the microscope and verified by PLA assay.
- During 2DG treatment, SG formation monitored using epitope tagged Pab1 and Pbp1 is impaired in lsm7 Δ background although PB formation was unaffected in lsm7 Δ background suggesting that role of Lsm7 in facilitating SG formation is not related to PB formation.
- Lsm7 overexpression enhanced SG formation.
- Internal deletions removing the aggregation prone domains in Lsm7 impair Lsm7 foci formation and SG formation. Similarly, fusion of spacer sequence from Sup35 to the C-terminus of Lsm7 decreased the formation of both Lsm7 foci and SGs.
- Time course study of Lsm7 and SG formation during 2DG treatment revealed that Lsm7 foci form well before SGs form and SGs stay bound to the side of Lsm7 foci.
- Colocalization data also revealed that Lsm7 foci contain RNA.
- In vitro, purified Lsm7 was able to undergo LLPS.
- Pre-treatment of cells with 1,6-hexanediol prevented SG and Lsm7 foci formation. However, treatment with 1,6-hexanediol after the Lsm7 foci and SGs are induced by 2DG, resulted in loss of only Lsm7 foci suggesting that Lsm7 foci are not needed to maintain SGs.

The observations are very interesting and overall, this is a very well carried out study. However, the manuscript can be strengthened by addressing some important questions especially about the Lsm7 foci which the authors argue is involved in initiating the Pab1 containing SGs.

The authors show that the Lsm7 foci contain RNA. However, what about other Lsms? Do the Lsm7 foci contain free Lsm7 or the entire Lsm1-7 complex?

1. We thank the reviewer for this helpful suggestion. We have addressed this comment by checking the co-localization of other known Lsm1-7/Pab1 complex components with Pab1. The results are shown in the new Figure 2e. The co-localization level of other Lsm1-7/Pab1 complex components with Pab1 are significantly lower than that of Lsm7. This indicates that some free Lsm7 associates with Pab1 outside of the Lsm1-7-complex context.

Importantly, is the ability of Lsm7 to get incorporated into the Lsm1-7 complex critical for foci formation and triggering SG formation? For example, if point mutations impairing the incorporation of Lsm7 into the Lsm1-7 complex are introduced into Lsm7, do Lsm7 foci and SGs still form under those conditions?

2. We agree with the reviewer, this would be a very meaningful experiment to do. Unfortunately, we could not find any reports of known point mutations that impair the incorporation of Lsm7 into the Lsm1-7-complex. One approach would have been to delete/impair the Lsm7 neighboring proteins in the heptameric ring, Lsm4 and Lsm5, however these are essential to cell survival. However, the high percentage of Lsm7 foci co-localizing with Pab1 granules, as compared to the other components of the complex (Figure 2e) indicates that Lsm7 partakes in SGs individually as well as part of the complex. Moreover, Lsm7 has been reported as one of the most stable proteins in the cell with a protein half-life above 100 h, much longer as compared to the other Lsm1-7/Pat1 complex components (9.1-16.7 h) (Christiano et al, 2014), indicating that Lsm7 might have other functions in addition to being a subunit of the Lsm1-7/Pat1 complex.

The authors show that knocking out PB components like Dcp2 impairs Lsm7 foci formation. Does this also happen in *lsm1Δ* background? If yes, under those conditions, do the fewer Lsm7 foci present still support SG formation?

3. Deletion of *LSM1* does indeed impact Lsm7 foci formation under 2-DG stress, as well as Pab1 granule formation (Figure 2f/g, Supplementary Fig. 2d). The Lsm7 foci that remain still co-localize with the Pab1 granules to a high degree (Supplementary Fig. 2d) suggesting a continued support function albeit hampered as seen in the reduction of Pab1 granules. In Supplementary Fig. 2e, we show the increased nuclear localization of Lsm7 upon *LSM1* deletion, indicating that Lsm1 influences the cellular localization and thereby cytoplasmic foci formation of Lsm7 under 2-DG treatment.

It is known that over expression of Lsm8 can sequester Lsm2 through Lsm7 proteins in the nucleus. Does that impair Lsm7 foci formation and SG formation?

4. We greatly appreciate this constructive suggestion, which led to additional evidence supporting the role of Lsm7 in SG regulation. We performed the overexpression experiment as suggested and the results are shown in the new Figure 2h. When overexpressing *LSM8* during 2-DG stress, there is a statistically significant decrease in both Lsm7 foci formation and Pab1 granule formation. The decrease in Lsm7 foci we see is in accordance with the findings of Spiller et al. 2007, where they showed that overexpression of *LSM8* caused an increase in the nuclear fraction of Lsm7. An increase in the nuclear localization of Lsm7, due to overexpression of *LSM8*, could explain the decreased ability of Lsm7 to form cytoplasmic foci and subsequent decrease in SG promotion that we observe.

Similarly, do point mutations in the conserved RNA binding loops of Lsm7 impair Lsm7 foci formation and SG formation?

5. We agree with the reviewer, this would be a relevant experiment to do. Unfortunately, although with a great effort, we still cannot find previous reports on such point mutations. We have only found a relevant paper by Zhou et al. 2014 (“Crystal structure and biochemical analysis of the heptameric Lsm1-7 complex”). In this paper the RNA binding affinity of the Lsm1-7 complex was not remarkably affected when replacing the conserved Arg (an invariant Arg residue shown to be important for RNA recognition by the Lsm2-8 complex) by Ala in Lsm7. Indicating that other components of the complex, such as Lsm2 and Lsm3, might have a bigger impact on RNA recognition and binding than Lsm7. However, that does not explain whether RNA binding is important in the context of Lsm7 foci formation and further SG promotion. However, the reviewer’s concern was addressed by the following data. 1, The high co-localization (88%) of Lsm7 foci with the SytoRNA probe indicates that Lsm7 foci do associate, if not directly at least indirectly, with RNA under 2-DG stress (Figure 4h). 2, The results in Supplementary Fig. 4d-f further support the notion that Lsm7 needs access to non-translating mRNAs to form foci. 3, the Lsm7 droplet formation can be enhanced by the addition of total RNA *in vitro* (Figure 6d). 4, Lsm7 seems to have a protective effect on Pab1 demixing *in vitro* upon total RNA addition (Figure 6e). Altogether, the presence of RNA seems to influence Lsm7 foci formation and even promote Lsm7 phase separation *in vitro*.

Addressing some of these questions will provide important insight into the nature of Lsm7 foci (and the mechanisms by which they facilitate SG formation) and make the story more complete.

The *in vitro* experiments are very interesting. However additional data to support their relevance to *in vivo* observations would help. For example, do the internal deletions in Lsm7 (that remove the aggregation prone domains) or fusion of spacer sequence to Lsm7 impair LLSP of Lsm7 *in vitro* also?

6. We fully agree with the reviewer and therefore decided to include the *lsm7 Δ IDR*-GFP and *lsm7 Δ 90-103*-GFP mutants in the *in vitro* setup together with WT Lsm7-GFP (Figure 5d). We used the *in vitro* conditions, 10% Dextran 70 and 5 μ M protein concentration, to detect the phase separation properties among the mutants. Under this condition, WT Lsm7 can still form phase separated droplets whereas both of the *lsm7* domain mutants display no phase separation at all (Figure 5d), in accordance with our *in vivo* foci formation data (Figure 4b).

Other questions/comments:

Fig1: What is the impact of 2DG treatment on the expression of epitope tagged Lsm7?

7. 2DG does not impact the expression of epitope tagged Lsm7, as can be seen in the new Figure 1e.

Fig2D: Western should also include Pbp1-GFP control

8. All data relating to Pbp1 has been added into its own figure (Supplementary Fig. 2). Neither *LSM7* deletion nor overexpression affects the protein expression of Pbp1 (Supplementary Fig. 2c).

Fig2E: How is PB formation induced and what do the time points (20 min and 120 min) refer to?

9. The PB formation was induced by adding 400 mM 2-DG and visualized 20 minutes and 120 minutes post 2-DG addition. We have now clarified this in the legend of Figure 3a.

Reviewer #2 (Remarks to the Author):

In this paper the authors describe a facilitating role for Lsm7 in the formation of 2-deoxyglucose induced stress granules in yeast, based on results from a prior published GFP library co-localization screen. They show during a 2DG stress timecourse that Lsm7 forms foci (aka seeds, scaffolds) prior to Pab1 following 2DG stress; Pab1 foci physically dock on the side of Lsm foci continue to enlarge whereas Lsm7 retain a certain size. Lsm7 is shown to phase separate in vitro, through these experiments do not really provide significant insight. A distinct sensitivity of Lsm7 and Pab1 foci to 1,6 hexanediol is observed (Lsm7 foci are sensitive, Pab1 foci are not), which reportedly affects only liquid-like assemblies, not solid assemblies – though it should be noted there are contradictory findings in the field about the use of this chemical, and how it should be interpreted.

While there are minor fixable issues with the data as presented, and some overstatements, the paper is relatively short and lacking in novelty, impact or mechanistic understanding that would warrant publication in a high impact journal. I have suggested new experiments, but the authors are probably best advised submitting elsewhere.

Major points:

- 1.) Clearly many granule assembly promoting proteins have been reported, and many identified that phase separate. Also, step-wise assembly of granule assembly has been reported in yeast and human models (e.g., Wheeler eLIFE 2016; Cirillo et al, Current Biology 2020). So there isn't a great deal of novelty here as of yet. A stronger justification as to why Lsm7 was chosen for follow up analysis in the screen would also help, since as a known PB protein, the degree of new insight one could gain already seems a potentially limited.

10. First of all, we would like to clarify that the GFP library co-localization screen data shown in this manuscript is a new data set that has never been published before.

Yes, we agree with the reviewer that “many granule assembly promoting proteins have been reported, and many identified that phase separate” and step-wise assembly of granule assembly has been reported under specific stress conditions (NaAsO₂ stress, Wheeler eLIFE 2016; NaN₃ and arsenite stress, Cirillo et al, Current Biology 2020). But

we think that our understanding of machineries linking protein phase separation and SG formation are still not complete. Especially previous studies have indicated that the components of SGs and the regulation mechanisms can be different under different stresses (reviewed in Campos-Melo et al., 2021). In line with this, we find *Lsm7* does not have an effect on SG formation induced by NaN3 (Figure 1d) as shown by Cirillo and colleagues.

With the substantial new data added in this revision, we hope that the reviewer might agree with us on that our results provide a potential new machinery on how *Lsm7* affects SG formation independent from its role in the *Lsm1-7/Pat1* complex in PBs. The possibility of direct mRNP remodeling functioning in SG formation machinery was also suggested in Buchan et al., JCB, 2008, Fig 9, dashed arrows). In this manuscript we provide a new set of data showing that, besides its function as a *Lsm1-7* complex/PB component, *Lsm7* can form liquid phase separation droplets and further initialize SG formation.

We have checked the original papers that have reported *Lsm1-7* as PB components. The first yeast PB paper (Sheth and Parker, 2003) use *Lsm1-GFP* (/RFP) as a reporter, the localization of other *Lsm* proteins were not included. In the Human PB paper (Ingelfinger et al., 2002), hLSm1 and hLSm4 antibodies were used for investigating the localization in PBs. These are very reasonable choices, since these studies focused on the *Lsm1-7*'s role in decapping in PBs, in which the formation of the complex is definitely required. But whether there are possible functions of each individual component on PB or SG formation were not evaluated.

In this study, we found an extra function of *Lsm7* involved in SG formation mechanism based on that *Lsm7* does not form foci under non-stress condition, indicating that it might not localize to PBs under such condition. Moreover, under 2-DG stress, the co-localization level of *Lsm7* with the SG marker (*Pab1*) is significantly higher than for the other *Lsm1-7* complex components. This indicates that besides its role as a *Lsm1-7* complex subunit, *Lsm7* itself might have an independent function in SG formation.

Please see the list below in support of the *Lsm7* independent function in SG formation, which also justified our choice of *Lsm7*.

- *Lsm7* has been reported as one of the most stable proteins in the cell with a protein half-life above 100 h, much longer as compared to the other *Lsm1-7/Pat1* complex components (9.1-16.7 h) (Christiano et al, 2014), indicating that *Lsm7* might have other functions in addition to being a subunit of the *Lsm1-7/Pat1* complex.
- *Lsm7* does not display the typical foci formation phenotype as PBs under unstressed conditions (Supplementary Fig. 3).
- The deletion of *LSM7* does not impact PBs (Figure 3a) but does result in decreased SG formation (Figure 2a).
- We overexpressed *LSM7* in an *EDC3* deletion mutants, and found that overexpression of *LSM7* can significantly increase SG formation, but do not have a significant effect on PB formation (Figure 3c).
- *Lsm7* foci do not grow in size during prolonged stress (Figure 4e and 4f) as reported for PBs (*Dcp2* as a marker, Teixeira et al., 2005).

- The independent effects of Lsm7 on SGs are further supported by the *in vitro* data showing that Lsm7 can form LLPS assemblies without the presence of other proteins (Figure 5) and that these condensates can co-phase separate with Pab1, and Lsm7 can enhance Pab1 droplet formation when added together (Figure 6b and 6c). Moreover, Lsm7 can protect Pab1 demixing upon addition of RNA *in vitro* without the addition of other proteins (Figure 6e).
- Lastly, the *in vivo* FRAP data indicates that Lsm7 foci are significantly faster at fully recovering after photobleaching, as compared to Dcp2 foci, indicating that Lsm7 foci are more dynamic in their recovery than Dcp2 foci (Figure 7d-f).

2.) As the authors are surely aware, in yeast, SGs and PBs spatially overlap under weak stress conditions, or in the early stages of some stress events. I rather suspect 2-DG, given the low number of foci/cell, and cells with foci, would show strong co-localization of Pab1 with Dcp2 or other PB markers. Have they ever checked this? If so, it would be useful to know if Lsm7 (being better known as a PB protein before this given it's function in the Pat1/Lsm complex), shows any foci and co-localization with PB markers in the absence of stress. i.e., add a no stress control to Figure 1D. This would change the perspective on Lsm7 uniquely seeding SGs, to the idea that this PB-localized protein seeds SGs (similar to what was suggested for PBs aiding -Glu SG assembly in Buchan et al, 2008).

11. We understand the reviewer's concern in this. We checked Lsm7 and Dcp2 foci formation and co-localization under no stress and 2-DG stress conditions (new Supplementary Fig 3a). In the absence of stress, Lsm7 does not form foci, contrary to Dcp2, and has therefore no foci co-localization with Dcp2. Dcp2 was mentioned in Figure 1b to have co-localization with Pab1 under 2-DG stress. We have highlighted this further in the new Supplementary Fig. 1a. Lastly, see Figure 2b (lower panel) for the unstressed Pab1 phenotype.

- Related to this point, Lsm7 is part of the Lsm complex. The authors should check other Lsm nulls to see if the same effects are seen or not. If not, then it at least suggests the possibility of effects beyond function in the Lsm complex. Note; Lsm4 is already implicated in PB assembly, and so this at least seems an obvious related factor to check for Pab1 effects in their 2DG system.

12. We thank the reviewer for this highly valid suggestion. The Pab1 2-DG stress phenotype for available null strains of the Lsm1-7/Pat1 complex components have been added to the new Figure 2f. The deletion mutants do not show the same effects as the *lsm7Δ* mutant in Pab1 granule formation (especially for *LSM2*, *LSM4* and *LSM6* mutants), suggesting that the *lsm7Δ* effects on SGs are not limited to Lsm7's function in the Lsm complex. However, we noticed that there is a statistically significant decrease in Pab1 granule formation in the *lsm1Δ* and *pat1Δ* strains as compared to the WT (*his3Δ*), and the Lsm7 foci formation were also reduced in the deletion strains (Supplementary Fig. 2d-e). The effects on SGs and PBs in the *pat1Δ* mutant have been reported before (Buchan and Parker, 2008). The great decrease in SGs in the *pat1Δ* mutant cannot be fully explained by its effect on PBs (minor decrease in PBs, Buchan and Parker, 2008),

indicating other mechanisms partaking in Pat1-related SG regulation. In line with this, we observed that deletion of either *PAT1* or *LSM1* results in a predominantly nuclear localization of Lsm7 under 2-DG stress, indicating that these mutants can affect Lsm7's cellular localization and subsequent Lsm7's ability to form cytoplasmic foci and further SG formation (Supplementary Fig. 2d).

Moreover, the double-mutant *lsm1Δ lsm7Δ* displays an even stronger loss of SGs (Figure 2g), as compared to the single mutants (Figure 2a and 2g), indicating an additive effect of Lsm7 on SG formation.

Furthermore, the effects of deletion of the C-terminal tail of Lsm4, which has a prion-type domain that contributes to aggregation of PBs (Decker et al., 2007), was tested. The *lsm4ΔC* strain does not show an impact on SG formation under 2-DG stress (Figure 2f). As mentioned by the reviewer, this strain has previously been shown to affect PB and SG formation under glucose depletion (Buchan and Parker, 2008). However, 2-DG is a weaker stressor, as also pointed out by the reviewer, resulting in glucose limitation not complete glucose depletion. It has been shown that the SG composition and regulation varies between different stressors (reviewed in Campos-Melo et al., 2021), then perhaps *lsm4ΔC*-related PB effects on SG formation require more severe stress conditions to emerge.

Minor points:

1.) As the authors acknowledge in the discussion, the observation of only 14 proteins in SGs is probably an artefact of their unusually high cut-off (60% GFP foci-co-localizing with Pab1 RFP). Since SG composition in yeast has something in the ballpark of >100 verified proteins (by candidate microscopy, and validation from SG purification studies e.g., Jain et al, 2016), it undermines confidence in the screen to not see so many known SG proteins in their list. What happens if you reduce the bar to something much smaller? This data could be included as a supplement; knowing the degree of localization of protein X in a SG below 60% would be informative.

13. We agree with the reviewer and have now provided a full list of proteins that co-localize with Pab1 granules (Supplementary Table 3). When including proteins below the 60 % co-localization threshold, the number of proteins exceeds 100.

2.) More detail on microscopy co-localization approach would be appreciated (there is mention of the software used only; in Fig1 legend, noted that co-localization was done manually – does that apply to the whole original screen, or just follow up testing?). Also, it is possible that some GFP-tagged proteins may be present in SG-distinct foci, implying other roles. Thus, it would be helpful if the authors did the opposite calculation to what they've done in 1B (i.e., divide number of GFP foci with RFP co-localization by total GFP foci).

14. The images and any foci co-localization were analyzed manually by using the composition function of the MetaXpress software. This has been clarified now in the Figure 1b legend and method section. In accordance with the reviewer's suggestion, we have included both types of calculations in Supplementary Table 3 (co-loc foci/total RFP foci and co-loc foci/total GFP foci).

3.) Fig S2: Based on previous Parker lab data, in which inactivating decapping enzymes (or

Xrn1) leads to strong increases in PBs, whereas Edc3 inactivation reduces PBs (presumably due to loss of scaffolding functions), it's slightly surprising to see Lsm7 foci decrease in both cases. Did they check PB phenotypes in these cell lines? Could they do a co-localization with a PB marker (e.g. Dcp1)? If Pb phenotypes in these nulls are as mentioned above, it would be further evidence that Lsm7 effects on SG formation are independent of PB effects (which is a point they make in Fig2).

15. As suggested by the reviewer, we have constructed *edc3Δ* and *dcp2-7* (TS) strains carrying tagged Lsm7 and PB marker (Dcp2-GFP or Edc3-RFP, respectively). We have also included the co-localization data as well (now Supplementary Fig. 3b and c). As shown before, we see a reduction in Lsm7 foci formation in both mutants as compared to WT. In both strains we also see a decrease in PB formation. Dcp2 plays a role in the assembly and/or maintenance of P-bodies, possibly through the multiple interactions between Dcp2p and other components of P-bodies (Teixeira and Parker, 2007). The complexity of Dcp2's role in PB regulation could render a possible explanation for the decrease in the PB formation phenotype we see in the *dcp2-7* (TS) strain. Here we use a temperature-sensitive allele strain together with 2-DG stress, in contrary to the previously reported PB phenotype in a complete *DCP2* deletion strain under glucose depletion (Teixeira and Parker, 2007). Differences in strain and stressor might influence the Dcp2 interaction complexity and subsequent PB phenotype.

There is a typo in regards to the *dcp2-7* TS strain in the previous version. This has now been corrected to clearly state that this is a temperature-sensitive allele, not a deletion (Supplementary Fig. 3c and Method section).

We note the reviewer's concerns regarding Lsm7's effects on SG formation independent of PB effects, and therefore provide a list of data in support of our claim in our No. 10 response to the first major point from the reviewer.

4.) The authors introduced 3 terms in the paper regarding domain signatures of Lsm7; aggregation prone domains (Fig 3A-B – they make truncations of these), IDRs and PLDs. They should make it clearer how (if at all) these overlap, and why they prioritize aggregation domains for analysis.

16. We agree with the reviewer that the reasoning for calling the aa 39-53 and aa 90-103 domains “aggregation prone domains” is somewhat misleading. Indeed, these domains also got highlighted by aggregation prone domain predictions but they are also hydrophobic regions. In regards to our story line it is more meaningful to describe that these are hydrophobic regions, due to such regions' implications in driving liquid-liquid phase separation, as well as protein aggregation (Sun et al 2008; Lin et al. 2021). We have decided to adjust accordingly in the manuscript (paragraph “Lsm7 foci formation is needed to promote SGs” and Methods). In addition to this, we decided to construct a *lsm7IDRA*-GFP Pab1-RFP strain, due to the emerging data on the importance of IDRs in protein phase separation (47–50) (Figure 4b). Figure 4a has been updated to include all domains mentioned and their positioning in regards to each other.

- Having shown that perturbation of Lsm7 affects SG foci, it would be prudent to check 1-2 other SG markers (e.g. Pbp1, Pub1, Ded1, eIF4G etc) to be sure that Lsm7 effects are not limited to only Pab1. I realize they do this for the null in 2B, but it's relatively little extra work.

17. In order to show that the effects of Lsm7 on SG formation are not limited to only Pab1, we have, in addition to the *LSM7* deletion data, also included *LSM7* OE data for Pbp1 under 2-DG stress condition (Supplementary Fig. 2a and b). The same phenotypic trends are true for Pbp1 as for Pab1. Moreover, we have also confirmed that the protein expression levels of Pbp1 were not influenced (Supplementary Fig. 2c).

5.) Figure 3C – the rationale to the C-terminal spacer tag experiment isn't exactly clear; i.e. why did the authors think this would impair Lsm7 function similar to a KO or truncation? Is the thought that having a charged sequence added may repel Lsm7 from interacting with itself.

18. The thought was that the charged M domain confers solubility to the yeast prion protein Sup35 and has been shown to dampen protein aggregation when fused to other proteins (Franzmann et al., 2018; Liu et al., 2002; Choe et al., 2016). Even though, as has been mentioned by the reviewers, the effects of this domain are complex, we wondered if, by fusing the spacer to Lsm7, it could alter the weak interactions required for LLPS and make the protein more soluble and block the foci formation of Lsm7. Thus, we constructed a Lsm7 strain carrying a spacer sequence (the charged middle domain (M) of Sup35) at the C-terminus of Lsm7-eGFP (Results showed in Figure 4c and d).

6.) 3E: Why not show data from the 4 biological replicates averaged?

19. Thanks for the suggestion from the reviewer, we have now shown the averaged data and SD for 4 replicates in Figure 4e.

7.) 3H should be quantified in terms of co-localization. Also might consider to oligo-dT instead of SYTO if specifically interested in mRNA.

20. Here we followed the suggestion from the reviewer and quantified the co-localization, which has been added to Figure 4h.

8.) The authors mention that the composition of Lsm foci is unclear. That could be assessed either unbiasedly (with Lsm7 purification coupled to prior centrifugation to select for higher molecular weight complexes + mass spec; akin to the SG purification steps in Jain et al), or with candidate approaches using reasonable guesses (e.g., other Lsm components, Pat1 etc). Making mutants in such components if they're established to be in the same foci, and assessing SG/PB effects would then give a little more insight into how SGs are being seeded. Note: The lsm4 delta c mutant data in Buchan et al 2008 isn't terribly different in terms of magnitude of effect compared to what is seen here; SG reduced about 2 fold, whereas PBs

largely unaffected.

21. We thank the reviewer for this constructive suggestion and we have looked at possible candidates based on reasonable guesses, as suggested by the reviewer. All candidates display some level of co-localization with Pab1 granules under 2-DG stress, but the co-localization levels are significantly lower than that of Lsm7 (Figure 2e), indicating that some free Lsm7 associates with SGs outside of the Lsm1-7-complex context. The Pab1 2-DG stress phenotype for available null strains of these candidates have been added to the new Figure 2f. Please see our response No. 12 for details.

9.) Respectfully, these types of single protein focused phase separation experiments don't reveal much in my view. Numerous proteins with IDRs, PLDs etc undergo phase separation at a suitably high concentration, with enough crowding agent, and basically all are impeded by high salt. It would be slightly more informative if other variables were introduced to this experiment such as:

- Lsm4 mutations (e.g. from 3B?) to establish which domains are key for phase separation properties.

22. We fully agree with the reviewer and therefore decided to include the *lsm7 Δ IDR*-GFP and *lsm7 Δ 90-103*-GFP mutants in the *in vitro* setup together with WT Lsm7-GFP (Figure 5d). We used the *in vitro* conditions, 10 % Dextran 70 and 5 μ M protein concentration, to detect the phase separation properties among the mutants. Under this condition, WT Lsm7 can still form phase separated droplets whereas both of the *lsm7* domain mutants display no phase separation at all (Figure 5d), in accordance with our *in vivo* foci formation data (Figure 4b).

- Presence/absence of RNA to establish if this influences ability to phase transition

23. We thank the reviewer for this suggestion and have now addressed this in Figure 6d and e. Lsm7 phase separation can be increased by the addition of total RNA (Figure 6d). During co-phase separation of Lsm7 and Pab1, in the presence of total RNA, Lsm7 can rescue the demixing of Pab1 (Figure 6e).

- Presence/absence of other SG proteins of interest. Can you recapitulate Pab1-Lsm7 docking phenomenon from 3F? Do you need the rest of the Lsm complex? Etc.

24. To address this concern from the reviewer, we decided to look at *in vitro* co-phase separation of Lsm7 and Pab1 (Figure 6b and c). When added together at the same concentration, Pab1 and Lsm7 co-phase separate into the same droplets (Figure 6b). Furthermore, the addition of 0.5 μ M Pab1 together with 5 μ M Lsm7 results in an increased droplet formation (Figure 6c), implying that co-phase separation of these two proteins can result in an enhanced effect, without the addition of other Lsm complex

components.

10.) Hexanediol is indeed used by a number of labs (Alberti and others) to make assertions about the liquid state of condensates. I suggest the authors at least consider (even better acknowledge) Wheeler et al, eLIFE 2016, which shows that hexanediol stresses yeast in numerous ways, and induces SGs in yeast. Why do the authors think they see something different than the Parker lab here? If they infact also see SG induction in response to hexanediol, might this complicate their data in Figure 5? Perhaps it would be worth checking if SGs are still impaired in assembly in an Lsm7 foci KO.

25. As mentioned by reviewer 2, others have indeed reported that 1,6-hexanediol can induce stress and subsequent stress granule formation under certain conditions and time-points (Wheeler et al., 2016). We have now addressed and acknowledged that in the manuscript paragraph “Lsm7 foci are liquid-liquid phase-separated condensates”. However, we would like to emphasize that in our setup (Figure 7a-c), we use different stress conditions (2-DG) than those used by Wheeler et al. 2016 (15 min glucose starvation). To make our data in Figure 7b clearer we have included images (Figure 7c) showing the dissolved foci phenotype we see when pretreating with 1,6-hexanediol+digitonin followed by 2-DG stress induction. Figure 7c shows that there is no SG induction due to the addition of 1,6-hexanediol. However, we can't rule out that there might be a slight additional SG (Pab1 granule) induction in the lower panel of Figure 7a, due to 1,6-hexanediol related stress. However, there is no statistically significant difference to the control (+digitonin) (Figure 7a, upper panel). In addition, the dissolving effects on Lsm7 are clear (Figure 7a, lower panel). This at least gives us a hint that the Lsm7 foci are potentially of a more liquid-like nature. We would like to stress that the 1,6-hexanediol assays are not used as conclusive methods to determine the *in vivo* physical states of Pab1 and Lsm7 foci, but more as an indicator. To further describe the phase separation nature of Pab1 and Lsm7 we have done *in vivo* FRAP showing Lsm7 foci to be more dynamic in their recovery as compared to Pab1 granules (Figure 7d-f), as well as *in vitro* phase separation assays displaying that domains needed for *in vivo* Lsm7 foci formation are also needed for *in vitro* phase separation (Figure 4b and 5d).

Other additional experimental suggestions:

a.) FRAP analysis of Lsm7 foci, in comparison to PB or SG markers (in vivo, maybe in vitro). Are there significantly different dynamics explaining distinct assembly/disassembly behaviors?

26. As suggested by the reviewer, we have performed *in vivo* FRAP analysis on Lsm7, Dcp2 and Pab1 foci under 2-DG stress (Figure 7d-f). Lsm7 foci display shorter half time rate and time needed for full recovery, as compared to Pab1, indicating a more dynamic nature (Figure 7f). Furthermore, Lsm7 foci have similar half time rates as Dcp2 foci; however, the selected Dcp2 foci display a big variance (Figure 7f, right). Nonetheless, the time Lsm7 foci require to fully recover after bleaching is significantly shorter than for Dcp2 foci, indicating faster dynamics (Figure 7d-f). The recovery rates for Pab1 and Dcp2 in our FRAP setup are overall faster than what has been shown before for Pab1

(heat shock), Dcp2 (unstressed,) and Lsm4 (PB marker, glucose depletion) (Zhu et al., 2020; Xing et al., 2020; Kroschwald et al., 2015), indicating varying foci dynamics under different stressors.

b.) Are Lsm7 foci dependent on translation repression? Do they still form if cells are previously treated with CHX prior to 2-DG treatment? Is their disassembly facilitated by CHX treatment? What about puromycin (same idea, opposite outcomes expected).

27. We appreciate these suggestions from the reviewer and have addressed them in new Supplementary Fig. 4d-f. When cells are pretreated with CHX, Lsm7 foci formation is significantly impinged under 2-DG stress (Supplementary Fig. 4d). Furthermore, the Lsm7 foci disassembly is facilitated by CHX (Supplementary Fig. 4e). In accordance with what has been shown for CHX effects (trapping mRNAs in polysomes) on stress granules (Buchan et al., 2008) and PBs (Sheth and Parker, 2003).

When treating a puromycin-sensitized strain with 2-DG+puromycin we see an increased induction in Lsm7 foci formation as compared to 2-DG treatment alone (Supplementary Fig. 4f). This implies that puromycin, through its ability to dissociate polysomes, can enhance Lsm7 foci formation under 2-DG stress, in accordance with what has been shown for PBs in yeast (Cary et al., 2014) and SGs (Kedersha et al., 2000).

These results, together with our *in vitro* RNA data (Figure 6d), indicate that the access to RNA affects Lsm7's ability to phase separate and form foci.

c.) Extend analyses to mammalian cells, in an interesting cell and stress context model.

28. We thank the reviewer for the suggestion of this very important future study. We hope that our results will raise the interest for groups with mammalian cells expertise. And indeed, as the reviewer suggested, by extending analyses to mammalian cells, in an interesting cell and stress context model (e.g. cancer related cell lines and conditions), will lead to deeper understanding of the potential role of Lsm7 on SG related oncogenesis or chemotherapy resistance machineries. This is one of the main reasons why we chose Lsm7, since it is a conserved protein from yeast to human. We hope that our results will lead to more extended studies in other models and generate a greater impact.

d.) Personally, I would be interested more as to why a factor like Iki3 (linked to RNA tRNA modification and RNAP II function) is in SGs. In any case, doing at least a null analysis (or over-expression?) and Pab1-RFP microscopy for all the hits previously not examined would give potentially yield greater insight as to undiscovered regulators of SG assembly. SG disassembly phenotypes could also be examined potentially with something like degron alleles.

29. We share the same interest as the reviewer pointed out. Currently, characterizing the role and mechanisms of Iki3 is an ongoing project. Since this manuscript focuses on Lsm7, we feel that extra data on Iki3 and other undiscovered regulators may not add extra supports and may deviate from the main focus on the role of Lsm7. Same for the SG disassembly phenotypes, detailed investigation on all factors identified from this

project will lead to another interesting project. We very much appreciate this suggestion on our future projects from the reviewer.

General narrative comments:

A few statements are overblown in the paper. Use of words like essential (e.g. line 257) and required (line 31, 86, 221, to described SG assembly phenotypes that while significant, are modest in fold change. These should be adjusted.

30. We agree with the reviewer and have modified accordingly and made changes on “essential” and “required” together with other necessary narrative adjustments.

Also, the authors claim in their abstract to identify Pab1-association proteins in their abstract (which I would take to mean physical interaction), and that Pab1 directly interacts with Lsm7. In fact, the only assay presented regarding Lsm7-Pab1 interaction is a proximity ligation assay (Fig 1E), which isn't direct proof of interaction (just close proximity). These claims should be adjusted, or experiments conducted (e.g., recombinant protein IPs, ideally with mutant Lsm7 constructs they went to the trouble of making) to show whether these proteins bind, and via what domains. Otherwise, even a standard IP would add something.

31. We have now adjusted the claims as suggested by the reviewer, using “Pab1 co-localized proteins” instead of “Pab1-association proteins”.

Reviewer #3 (Remarks to the Author):

In this manuscript, the authors identify Lsm7 as a key early component of 2-deoxyglucose-induced stress granules. Although many components and regulators of stress granules have been identified, Lsm7's early (preceding Pab1 condensation) and almost-essential role in stress granule nucleation make this finding particularly interesting. Overall, the experiments are well done and clearly described. The authors nicely establish that although Lsm7 is important for stress granule formation, it is dispensable for the ongoing stability of these assemblies. Nevertheless, there are a few issues that should be addressed.

1. It would strengthen the manuscript to test whether Lsm7 plays a role in SGs induced by other stresses. Either result would be interesting, but such experiments would provide insight into Lsm7's mechanism of action, revealing whether its assembly is a specific response to 2-deoxyglucose-induced stress, or a more generic stress response.

32. We thank the reviewer for this highly relevant suggestion and we have now added Figure 2d showing the impact of *LSM7* deletion on SG formation under 4 other stress conditions. Lsm7 also impacts SG formation under glucose starvation (-glu 2 h) and stationary phase, but not during sodium azide stress (NaN₃) and heat shock, indicating that Lsm7 plays a greater role during nutrient and glucose stress responses (Figure 2d).

2. The authors consistently quantify SG and P-body formation based on the fraction of cells

with foci. However, this would seem to provide an incomplete picture, as many of the mutations seem to affect not only the number of foci, but also the intensity of these foci. Thus, it seems important to quantify both the number of cells with foci, and the fraction of signal in these foci. This is particularly important in cases where the authors are arguing that a mutation does not impact assembly. For example, the authors argue that *Lsm7* deletion does not reduce P bodies; however, by eye it appears that although most cells still have P bodies, they are on average less intense.

33. We thank the reviewer for this suggestion and we have now quantified the signal intensity in the instances where a mutation does not seem to affect the number of foci. The relative Dcp2-GFP signal intensity of the *lsm7Δ* mutant did not differ from the WT (Figure 3b).

3. For *Lsm7* overexpression (Fig 2C), it is important to show cells both before and after stress, to ensure that overexpression isn't inducing assembly independent of stress.

34. We agree with the reviewer and have now added the before stress data (Figure 2b, lower panel "unstressed"). Overexpression of *LSM7* does not induce Pab1 granule formation in the absence of stress.

4. In Figure 3, the authors should confirm that their mutations (deletion of aggregation-prone segments and addition of the M domain) don't impact expression, as they have already shown that *Lsm7* expression can influence SG formation.

35. We thank the reviewer for this suggestion and we have added this data in Supplementary Fig. 4b. The introduced *LSM7* mutations or addition of the M domain (spacer) do not impact the expression level of *Lsm7*.

5. In Figure 3, the authors fuse the Sup35 M domain to their construct to prevent assembly. While this indeed worked, the authors give little rationale for choosing this domain. The M domain is a regulator of stress-induced assembly (Franzmann et al, 2018). While it prevents formation of irreversible aggregates by Sup35, it promotes formation of gels in response to stress. Given this complex relationship with protein assembly, it is a surprising choice for a solubilization domain, so a more thorough explanation for this choice would be helpful.

36. The thought was that the charged M domain confers solubility to the yeast prion protein Sup35 and has been shown to dampen protein aggregation when fused to other proteins (Franzmann et al., 2018; Liu et al., 2002; Choe et al., 2016). Even though, as has been mentioned by the reviewers, the effects of this domain are complex, we wondered if, by fusing the spacer to *Lsm7*, it could alter the weak interactions required for LLPS and make the protein more soluble. Thus, we constructed a *Lsm7* strain carrying a spacer sequence (the charged middle domain (M) of Sup35) at the C-terminus of *Lsm7*-eGFP (Results showed in Figure 4c and d).

6. The authors propose multiple models by which Lsm7 could initiate stress granule formation, including trapping RNA that facilitates Pab1 binding, or directly interaction with Pab1. It is therefore surprising that the authors don't test include Pab1 and RNA in their *in vitro* experiments, to see if Lsm7 promotes Pab1 assembly, and whether RNA affect either Lsm7 assembly or Pab1/Lsm7 co-assembly.

37. We appreciate the reviewer for these suggestions, which have also been brought up by multiple reviewers. We have now included both Pab1 and addition of RNA in our *in vitro* setup. Both purified Pab1 and Lsm7 phase separate *in vitro* (Figure 6a). When added together at the same concentration, Pab1 and Lsm7 co-phase separate into the same droplets (Figure 6b). Furthermore, the addition of 0.5 μ M Pab1 together with 5 μ M Lsm7 results in an increased droplet formation (Figure 6c), implying that co-phase separation of these two proteins can result in an enhanced effect. Furthermore, Lsm7 phase separation can be increased by the addition of total RNA (Figure 6d). Addition of total RNA to Pab1 alone results in loss of demixing (Figure 6d), in accordance with what has been shown *in vitro* before (Riback et al., 2017). Lastly, during co-phase separation of Lsm7 and Pab1, addition of Lsm7 can rescue the phase separation of Pab1 in the presence of total RNA (Figure 6e).

7. The authors repeatedly describe Lsm7 as “essential” for 2-deoxyglucose-induced SG formation (including in the abstract), yet according to their data (Fig 2), Lsm7 deletion reduces, but does not eliminate SG formation. Thus, while it is clearly important, it is not accurate to say that Lsm7 is “essential.”

38. As suggested by the reviewer, we have adjusted our expression accordingly.

REVIEWERS' COMMENTS

Reviewer #1 (Remarks to the Author):

The revised version of the manuscript is certainly much improved compared to the original version with a more complete story. The authors have satisfactorily addressed the comments by including additional data that clarifies how their observations relate to the function of the Lsm1-7-Pat1 complex.

Reviewer #2 (Remarks to the Author):

Although the magnitude of effect of Lsm7 on stress granule formation in yeast following 2DG remains somewhat modest, I must commend the authors for addressing the vast majority of my comments in a very thorough manner. The number of new experiments, and responsiveness to writing/data presentation suggestions was significant. The key improvements are as follows:

- Clearer identification of a PB (and Lsm complex)-independent function for Lsm7 in SG assembly (and better rationale explanation for examining it in the first place)
- Significant improvements to the in vitro LLPS data showing (interestingly) somewhat distinct Pab1-Lsm7 co-phase separation behavior (unlike docking phenotype in vivo). Maybe this should be commented on a little further in the discussion? Regardless, the impacts of Lsm7 mutants and RNA are a welcome addition also to this dataset.
- Useful FRAP data assessing dynamics of Lsm7 foci in relation to other SG and PB markers

I also apologize for the confusion on my end regarding the screen - I had confused this with a prior Sunnerhagen lab screen done with 2-DG (which was looking for genes affecting SG and PB-assembly - PLOS Genet 2014; Yang et al). Out of curiosity, did Lsm7 show up here? If so, probably worth a mention.

Overall, I believe the paper is significantly improved and am happy to recommend publication. The authors have addressed essentially all my previous comments, with the exception of future experimental suggestions which I understand are beyond the scope of the work here.

The two suggestions I made above (Discussing Lsm7/Pab1 differences in co-LLPS in vitro and in vivo, and possible links back to prior 2-DG screen) are optional, and I leave for the authors to decide if they wish to address.

Reviewer #3 (Remarks to the Author):

The authors have appropriately addressed my concerns from the previous submission, and in my opinion have nicely addressed issues raised by the other reviewers. I believe this work represents an important contribution to the field.

REVIEWERS' COMMENTS

Reviewer #1 (Remarks to the Author):

The revised version of the manuscript is certainly much improved compared to the original version with a more complete story. The authors have satisfactorily addressed the comments by including additional data that clarifies how their observations relate to the function of the Lsm1-7-Pat1 complex.

We thank the reviewer for their helpful comments and suggestions that facilitated us in making our manuscript stronger.

Reviewer #2 (Remarks to the Author):

Although the magnitude of effect of Lsm7 on stress granule formation in yeast following 2DG remains somewhat modest, I must commend the authors for addressing the vast majority of my comments in a very thorough manner. The number of new experiments, and responsiveness to writing/data presentation suggestions was significant. The key improvements are as follows:

- Clearer identification of a PB (and Lsm complex)-independent function for Lsm7 in SG assembly (and better rationale explanation for examining it in the first place)
- Significant improvements to the in vitro LLPS data showing (interestingly) somewhat distinct Pab1-Lsm7 co-phase separation behavior (unlike docking phenotype in vivo). Maybe this should be commented on a little further in the discussion? Regardless, the impacts of Lsm7 mutants and RNA are a welcome addition also to this dataset.
- Useful FRAP data assessing dynamics of Lsm7 foci in relation to other SG and PB markers

I also apologize for the confusion on my end regarding the screen - I had confused this with a prior Sunnerhagen lab screen done with 2-DG (which was looking for genes affecting SG and PB-assembly - PLOS Genet 2014; Yang et al). Out of curiosity, did Lsm7 show up here? If so, probably worth a mention.

Overall, I believe the paper is significantly improved and am happy to recommend publication. The authors have addressed essentially all my previous comments, with the exception of future experimental suggestions which I understand are beyond the scope of the work here.

The two suggestions I made above (Discussing Lsm7/Pab1 differences in co-LLPS in vitro and in vivo, and possible links back to prior 2-DG screen) are optional, and I leave for the authors

to decide if they wish to address.

Before addressing the two suggestions made here by the reviewer, we would like to thank the reviewer for the insightful comments and suggestions that contributed to a stronger manuscript.

We agree with the reviewer and decided to mention the need for further studies into the link between the *in vivo* docking and *in vitro* co-phase separation phenotypes of Pab1 and Lsm7 in the discussion.

LSM7 was not among the hits in the screen published by Sunnerhagen lab (PLOS Genet 2014; Yang et al.). It was later established that it was a false negative hit due to variance in the replicates and was confirmed afterwards as a true hit with manual confirmation in this work. To avoid potential confusion or unnecessary distraction, we prefer to not discuss this issue in the manuscript.

Reviewer #3 (Remarks to the Author):

The authors have appropriately addressed my concerns from the previous submission, and in my opinion have nicely addressed issues raised by the other reviewers. I believe this work represents an important contribution to the field.

We thank the reviewer for their valuable comments and suggestions which strengthened our manuscript.